# Machine Learning Methods for Multiscale Physics and Urban Engineering Problems

**DOI:** 10.3390/e24081134

**Published:** 2022-08-16

**Authors:** Somya Sharma, Marten Thompson, Debra Laefer, Michael Lawler, Kevin McIlhany, Olivier Pauluis, Dallas R. Trinkle, Snigdhansu Chatterjee

**Affiliations:** 1Department of Computer Science and Engineering, University of Minnesota-Twin Cities, 200 Union Street SE, Minneapolis, MN 55455, USA; 2School of Statistics, University of Minnesota-Twin Cities, 313 Ford Hall, 224 Church St SE, Minneapolis, MN 55455, USA; 3Department of Civil and Urban Engineering, New York University, Rogers Hall RH 411, Brooklyn, NY 11201, USA; 4Department of Physics, Applied Physics and Astronomy, Binghamton University, 4400 Vestal Parkway East, Binghamton, NY 13902, USA; 5Physics Department, United States Naval Academy, 572 Holloway Rd. m/s 9c, Annapolis, MD 21402, USA; 6Courant Institute of Mathematical Sciences, New York University, 251 Mercer Street, New York, NY 10012, USA; 7Department of Materials Science & Engineering, University of Illinois, 201 Materials Science and Engineering Building, 1304 W. Green St. MC 246, Urbana, IL 61801, USA

**Keywords:** spin ice, approximate Hamiltonian, molecular dynamics, moist atmosphere dynamics, urban engineering, hybrid approach, time evolution, dimension reduction, multi-resolution Gaussian Process, approximate Bayesian computation

## Abstract

We present an overview of four challenging research areas in multiscale physics and engineering as well as four data science topics that may be developed for addressing these challenges. We focus on multiscale spatiotemporal problems in light of the importance of understanding the accompanying scientific processes and engineering ideas, where “multiscale” refers to concurrent, non-trivial and coupled models over scales separated by orders of magnitude in either space, time, energy, momenta, or any other relevant parameter. Specifically, we consider problems where the data may be obtained at various resolutions; analyzing such data and constructing coupled models led to open research questions in various applications of data science. Numeric studies are reported for one of the data science techniques discussed here for illustration, namely, on approximate Bayesian computations.

## 1. Introduction

The topic of modeling spatiotemporal physical processes using scientific and data-driven techniques is vast, and its scope extends to countless topics in physical and natural sciences and related engineering domains. Within this vast space, studying predictive and scientific models for multiscale phenomena and for problems where the data may be obtained and modeled in a number of resolutions or granularity is of particular interest. This concept of multiscale, mixed-resolution problems is interesting, because many physical phenomena naturally exhibit multiscale behavior and many engineering applications require understanding and leveraging features at multiple resolutions. Furthermore, technological, budgetary, ethical and other considerations may limit the quantity of data that can be gathered at specific resolutions. Thus, there is a need to study predictive inference and modeling of mixed-resolution data and multiscale or mixed-scale processes. To complement this, modeling mixed-resolution data and multiscale phenomena often requires the development of new data science techniques.

To avoid confusion, the concepts of *multiscale* phenomena and *mixed-resolution* data require disambiguation. Some physics phenomena are observed and studied on distance scales of centimeters–meters and on time scales of milliseconds to hours. This may be considered by many to be a large/long distance scale (macro). By comparison, the lower (and hard) limit in physics is at the quantum scale, meaning that distances of nanometers and times from nano- to pico-seconds, making this the smallest scale (micro). Many problems further require understanding at a meso-scale (somewhere in between—often at micrometer–millimeter sizes and times from sub-microseconds to sub-milliseconds). Meso-scale examples are found readily in magnetism and polarization models—often exploited for use in liquid crystal technologies, as well as solid-state information storage or retrieval. Similar in scope (micro/meso/macro), but different in nature, are problems related to chemistry and spectroscopy where the relevant scales are based on energy/momentum and frequencies/wavelengths. Further, there are other scientific domains where some relevant parameter exhibits multiscale behavior separated by orders of magnitude.

Recognizing that scientific and engineering domains currently leverage features spanning large differences in scope, the computational approaches adopted to address these problems require *multiresolutional* grids or other schemes that can computationally scale in terms of the storage required and the CPU/GPU/APU operations needed. As a simple example, when modeling some of Earth’s features such as the depth of the water table or the extent of forest cover or urbanization from the surface to near space, models can have resolutions of several meters up to one kilometer. As we increase the elevation, models may have resolutions starting at one kilometer and range up to tens of kilometers. Finally, as we approach the upper atmosphere and into space, the models once again need a resolution shift—now in the hundreds of kilometers range. The demarcation in each case is a radial distance from the center of the Earth. At a given radius, the values from one model need to match boundaries to a differing model, where the values from the finer grid are usually averaged over onto the coarser grid. However, this information exchange is two-way, thus, the values from the coarser model also need to be fed into the boundary conditions of the finer grid model; this often requires an interpolating function to be employed to provide a reasonable continuity at a given radius. Artificial neural networks and other machine learning (ML) techniques excel at interpolation/extrapolation, making them good candidates to employ to address this “mixed-resolution” issue. As more effort is being placed on integrating and understanding the limitations of ML techniques, increased adoption of these techniques within traditional sciences and engineering has started. Throughout this paper, a recurring theme is that of computational science and engineering (CSE) today becoming driven less by raw computational resources such as CPU/GPU/PFLOPS and focusing more on *how* large amounts of data can be leveraged to improve accuracy (spatially) and prediction (temporally). Historically, exploration of many fields was limited by CPU/clock speed and available memory, with numerical techniques developed to process either simulations or data in the most *efficient* manner due to the absence of alternative approaches.

In this context, this paper aims to provide overviews of two things. First, we present discussions and a contextual review of a number of physical and engineering problems where multiscale phenomena are exhibited and data are typically obtained or modeled at different resolutions. Then, we discuss some interesting ML techniques that are applicable to the introduced physics and engineering problems. In both the physics-related and ML-related parts of this paper, the topics for discussion and review have been carefully chosen to reflect a scope of open problems that currently exist that should be of interest to many researchers in these topics. Section 2, Section 3, Section 4 and Section 5 of this paper discuss several physics and engineering problems, followed by discussions on data science-related techniques in Section 6, Section 7, Section 8 and Section 9.

In particular, we include new data experiments using approximate Bayesian computational (ABC) techniques in Section 9.4. There, we study several properties of ABC, including the choice of prior, and the choice of sampling algorithm, distance metric, summary statistics, the choice of the tolerance parameter and the ABC sampling size. The effect of the sample size and noise variance on the performance of the ABC algorithm is also studied. In addition, we also take a deep dive into the efficiencies that may be achieved with ABC using parallel computational techniques. The quality of ABC outputs are studied in terms of the approximation error resulting from using a tolerance parameter, the information loss from using a non-sufficient summary statistic and the Monte Carlo error arising from using a finite number of ABC samples. We also replicate a study on using ABC techniques in population genetics and ancestral inference, and for this study, we have used some modern computational schemes as well.

Specifically, in Section 2, we present an overview of the long-time-scale, glass-like dynamics of spin ice and discuss whether these dynamics can arise purely from Pauling’s atomic scale ice rules. Then, in Section 3, approximate Hamiltonians for molecular dynamics are reviewed and discussed. This is followed in Section 4 with a discussion on the dynamics of moist atmosphere. The final section on domain science, Section 5, presents an overview of multiscale phenomena of interest and mixed-resolution data aspects in an urban engineering documentation context. We then begin in Section 6 the discussion of data science techniques that may be applied to the different physics and engineering challenges. We remark on multiple hybrid approaches that use scientific knowledge and ML techniques simultaneously. This section also offers a broad overview and classification of ML techniques that may be available to the scientist interested in exploring data science tools. Following this, in Section 7, we present a broad overview and discussion of dimension reduction techniques. For modeling multiscale processes and data of mixed-resolution, stochastic systems that can jointly model across scales are important. To this end, in Section 8, we present a brief overview of the multiresolution Gaussian process that can be an essential building block of data science models for multiscale processes. Finally, in Section 9, we address computational aspects by reviewing and discussing approximate Bayesian computations (ABC) in considerable detail, as a viable and efficient alternative to extremely computationally demanding techniques such as Markov Chain Monte Carlo (MCMC) or artificial neural networks (ANNs). In this section, we also provide some numeric examples to illustrate the computational strengths and potential weaknesses of ABC. Numeric examples are limited only to Section 9 to keep this paper of a reasonable length. Concluding remarks are presented in Section 10.

## 2. The Mysteriously Long Time Dynamics of Spin Ice

### 2.1. Ice Is only One-Third Frozen

Not all solids are completely “frozen”. Of course many are—the atoms in crystalline solids typically oscillate back and forth about an equilibrium position. Nearly all stay in place. Yet others, such as glass, can form an amorphous state where atoms take random positions and move slowly. These are not quite solid and flow like a supercooled highly viscous liquid, at a rate depending on their temperature [1]. This is a perplexing phenomenon that nevertheless does not explain the thickness of glass windows in European cathedrals [2]. Why glass flows is one of the oldest unsolved problems in physics [3] and, thus, ripe for new insights that might be produced by ML methods.

Remarkably, there is a middle ground between crystalline and glass solids in which apparently solid material have atoms that can move around within a crystalline cage. A solid-state lithium ion battery [4], such as those that may underlie a revolution in battery technology for electric cars [5], is a well-recognized example of this. The most common solid with this property is ice.

Water ice is fascinating from the above perspective. The hydrogen atoms in water ice constantly compete with each other over which ones bind to a specific oxygen atom. At any given moment, each oxygen atom binds with two hydrogen atoms to form an H${}_{2}$O water molecule. However, there are many ways for this to happen given a crystalline arrangement of oxygen atoms, such as in the diamond lattice shown in Figure 1, left panel. Here, each oxygen atom has four neighboring oxygen atoms just like carbon atoms in a diamond. Between any two oxygen atoms lives a hydrogen atom. Yet, the chemistry of the hydrogen–oxygen bond prevents the hydrogen atom from being equally shared between the two oxygen atoms. The hydrogen atom must be close to only one of the oxygen atoms. However, additionally, only two hydrogen atoms can be close to any one oxygen atom. From an energy perspective, an oxygen atom hosting one hydrogen atom or three is costly. Consequently, the hydrogen atom placement must obey Pauling’s ice rules [6]: two hydrogen atoms close to an oxygen atom and two far from it; often referred to as the “two-in-two-out” rule.

Physicists would describe the hydrogen atoms in water as “frustrated” due their surfeit of choices. When this happens, matter often cycles through many phases when changing its environment such as applying pressure and changing its temperature. Ice is a striking example of this phenomenon. In ice, while these phases are all solids, they have different crystalline ordering patterns or occasionally an amorphous phase. A sketch of water’s phase diagram is presented in Figure 1 right panel (for more details see Ref. [7]). The frustration of the hydrogen atoms leads to 19 distinct phases of ice [8].

The frustration phenomenon could be related to that of glass formation, with the slowly changing random arrangement of atoms in an amorphous solid akin to the abundant choices enabled by the ice rules [9]. If so, the complex interactions between atoms that produce the amorphous solid phase could be replaced with simpler rules. A mild attempt to do so called the “constraint theory of glass” [10] translates the complexity of atomic interactions for a set of constraints. Herein, we will take an even simpler approach than attempting a direct solution to the glass problem by proposing an interpretable model. Specifically, we will review studies of spin ice, a simpler system than water ice that nevertheless obeys Pauling’s ice rules and look for unexpectedly long time dynamics with clues to the mystery of glass.

### 2.2. Spin Ice: An Idealized Ice

Several aspects of water ice motivate the search for a more simple ice-like system. Water ice is complicated, because not only do the hydrogen atoms have to obey the ice rules, but they must do so while also living with oxygen atoms that are constantly vibrating due to a finite temperature. Fortunately, there is an analogous system to ice—spin ice materials [11]—where we can ignore this complication.

In Dy${}_{2}$Ti${}_{2}$O${}_{7}$, the Dy atoms live on a pyrochlore lattice, as shown in Figure 2 left. This lattice consists of tetrahedra whose centers define a diamond lattice (oxygen atoms live at these points in ice I${}_{c}$, see Figure 1) and whose vertices hold Dy atoms (shown in blue) and are located on the line connecting two tetrahedral centers. The Dy atoms behave like an atomic-sized bar magnet that must point towards one of the two neighboring diamond lattice sites (see Figure 2 right). Hence, the configurations of the magnetic moment of the Dy atoms are in one-to-one correspondence with the location of the Hydrogen atoms in water ice. Dy${}_{2}$Ti${}_{2}$O${}_{7}$ is, therefore, a magnetic, or “spin”, ice. Spin ice is a phase of matter proposed by Anderson in 1956 [12], but it was not discovered until 1999 [13], shortly after hints of it were observed in Ho${}_{2}$Ti${}_{2}$O${}_{7}$ [14].

A model for spin ice can be expressed in several ways that are amenable for simulations and ML techniques. From a lattice perspective, we can define the location of each Dy atom by identifying to which primitive face centered cubic unit cell (labeled by integers $n_{1}$, $n_{2}$, and $n_{3}$) and to which Dy site within this unit cell α∈{1,2,3,4} it belongs (together with the face-centered cubic Bravais lattice vectors $a_{i}$, i∈{1,2,3} and pyrochlore lattice basis vectors $d_{\alpha }$, α∈{1,2,3,4}, each atom is given an explicit location $r\left[n_{1},n_{2},n_{3},\alpha \right]=\sum_{i}n_{i}a_{i}+d_{\alpha }$, see Ashcroft and Mermin [16] Chapter 4 for an accessible discussion of crystallography). As a result, we can characterize each configuration of bar magnets by the binary tensor $S[n_{1},n_{2},n_{3},\alpha ]$, a three-dimensional (3D) grid with four channels similar to that of an image (which is two-dimensional with three channels, one per color). The equilibrium probability of obtaining any single configuration is given by the Boltzmann distribution
(1)$$P\left(S\right)=\frac{1}{Z}e^{-\beta E(S)}$$
where E(S) is the energy of one configuration (see below), $\beta =1/k_{B}T$ is the temperature parameter and *Z* is a normalizing constant. Equilibrium properties are then studied via a dataset of samples from this distribution using a Metropolis–Hastings Monte Carlo approach via an ML method [17]. The use of approximate Bayesian computations, discussed in Section 9 below, remains an open challenge.

To model the changes of configurations over time and to study the potential novel dynamics arising from ice rules, we need to model basic dynamical processes. Perhaps the simplest approach is to use a master equation:(2)$$\frac{d}{dt}P\left(S\backslash t\right)=\sum_{T}\left(R(S,T)P(T\backslash t)-R(T,S)P(S\backslash t)\right)$$
with rates
(3)$$R\left(S,T\right)=\{\begin{array}{cc} r_{0}e^{-\beta (E(S)-E(T))/2} & T=S+\mathrm{one}\;\mathrm{spin}\;\mathrm{flip}\\ 0 & \mathrm{otherwise} \end{array}$$
where $r_{0}$ is the characteristic rate of a single spin flip process and the magnetic moment of a Dy atom changes direction. This dynamical equation guarantees $\lim_{t\to \infty }P\left(S\backslash t\right)=P\left(S\right)$ by obeying the principle of detailed balance. We can then solve this equation using a kinetic Monte Carlo [18] approach and generate a dataset of spin configurations *S* over time.

To understand the affiliated energy, it is useful to express a spin configuration as a graph, instead of as a 3D grid $S[n_{1},n_{2},n_{3},\alpha ]$. To construct the graph, we consider the diamond lattice with the same unit cell as the Dy lattice and whose lattice points are placed at the centers of the tetrahedra of the Dy lattice. We denote these diamond lattice sites within the unit cell by index a=1,2. We can then form the vertex set *V* of the graph by indexing each of these sites via $i=n1+n2*N_{1}+n3*N_{1}*N_{2}+a*N_{1}*N_{2}*N_{3}$ for a system with $N_{1}*N_{2}*N_{3}$ unit cells. A configuration is then given by directed edges between nearest neighbor diamond lattice sites. We can denote the edges by an adjacency matrix $A_{ij}$ whose entries are 1 (if ij denotes a nearest neighbor bond and the Dy atom’s spin points from *i* to *j*) and 0 (otherwise). Physicists prefer to work instead with what amounts to a signed adjacency matrix $\sigma_{ij}$ whose entries are 1 (if there is a directed edge from *i* to *j*), −1 (if there is a directed edge from *j* to *i*), or zero (otherwise). The matrix σ is then antisymmetric, but there is a one-to-one correspondence between σ and the tensor *S*. In terms of σ, the simplest model for the energy E(S)≡E(σ) is given by
(4)$$E_{\mathrm{local}}\left(\sigma \right)=\frac{J}{4}\sum_{i}{\left(\sum_{j}\sigma_{ij}\right)}^{2}$$
where $J/k_{B}$ is an energy scale of about 1 kelvin. The set of ground states given by $\sum_{j}\sigma_{ij}=0$
*obey the ice rules*: two of the four edges point towards *i* and two point away from *i*.

A more realistic model would include the long-range, dipole–dipole energy between any two Dy atoms [19]. This additional contribution preserves the ice rules embodied in Equation (Equation 4), except at temperatures well below 1 kelvin. The zero energy configurations of Equation (Equation 4) and the low energy configurations of a more realistic model both contain the same set of configurations. However, the more realistic model is needed to describe violations of the ice rules known as “monopoles”.

### 2.3. Magnetic Monopoles and Transitions between Ice States

The model of spin ice dynamics in Equation (Equation 2) is characterized by random spin flip processes. Suppose a spin ice is momentarily obeying the ice rules. The graph describing its current configuration then has two edges directed into each vertex and two directed outwards. After one spin flip, one of these edges will point in the other direction. Suddenly, we have one vertex with three outgoing and one incoming edge and another vertex with three incoming and one outgoing vertex—two violations of the ice rules (see the visualization in the spin language presented in Figure 3). The probability of this happening is proportional to $e^{-\beta J}$ and so strongly suppressed for $T\ll J/k_{B}=1$ kelvin. However, such violations will necessarily happen if an ice rule obeying configuration were to change over time.

When the next spin flip process happens, several possibilities could occur. One possibility is that the same spin flips a second time and the ice rules are once again obeyed. Another possibility is that another spin flips far away from the first and generates another pair of ice-rule violating vertices. Perhaps, however, the most interesting process that could arise is for a spin to flip that neighbors the first spin. Then, either three sites violate the ice-rules with the middle site having a double violation, or only two sites violate the ice-rules, with the middle site preserving the ice rules. One of the violations then has “hopped” to a neighboring site but the total number of violations remains the same.

A beautiful feature of the graph description of a spin ice configuration is the visualization it enables. The directed edges express the magnetic field produced by the dipoles of each Dy atom. The visualization of the ice rule violation presented in Figure 3 then shows that the violation corresponds to creating a source or a sink for the magnetic field. The violation of the ice rules is, therefore, even worse than it appears, as it seems to violate Maxwell’s equations of electrodynamics, which dictate that magnetic fields do not have sources and sinks. Only electric fields have these, and they are charges. According to Maxwell’s equations, the only way to produce a magnetic field, to “source" a magnetic field, is to have a moving electric charge that generates a current. Thus, the visualization supplies us with insight into the magnetism of spin ice and the meaning to their violation of ice rules.

Fortunately, Maxwell’s equations are not actually violated. It only appears so inside Dy${}_{2}$Ti${}_{2}$O${}_{7}$. Maxwell’s equation *in a medium* involves two magnetic fields B and H with the Maxwell equation ∇·B=0 imposing the no-source-or-sink rule. However, $H=B/\mu_{0}-M$, with M the magnetization of the material, is free to violate this rule. Even though this is possible, until the discovery of spin ice, no material had been found that even appeared to violate Maxwell’s equations by having an H with a source or sink.

Dirac noticed that a violation of the no-source-or-sink rule would imply that there were particles that carry magnetic charge known as magnetic monopoles [21]. The mere existence of one of these particles could explain the quantization of charge (that all particles, such as protons, electrons and muons, carry an integer amount of the charge *e* of an electron). There is a long history of the search for such magnetic monopoles with no success to date. Yet, the violations of the ice rules inside Dy${}_{2}$Ti${}_{2}$O${}_{7}$ behave exactly as though they are these magnetic monopoles. Dy${}_{2}$Ti${}_{2}$O${}_{7}$ is a universe where such magnetic monopoles are commonplace.

The dynamics of spin ice is then intimately tied to the behavior of magnetic monopoles. Spin configurations transition from one to another through the production, hopping and annihilation of magnetic monopoles. It is, therefore, of great interest to know whether these monopoles hop around and form a gas of charges known as a plasma or whether they are slow and plodding and somehow characterize the motion of a glass.

### 2.4. Heat Capacity and the Existence of Spin Ice in Dy${}_{2}$Ti${}_{2}$O${}_{7}$

Notably, the ice rules also seemingly violate another fundamental theory of physics: the third law of thermodynamics. This law states that the entropy, which is the logarithm of the number of accessible configurations of a statistical mechanical system, vanishes as the temperature is brought to zero. The discovery that this law appeared to be violated provided the first evidence for the existence of spin ice. The amount the law is violated, namely the non-zero entropy at low temperatures, was estimated by Pauling in 1935 [6] and shown to agree with heat capacity experiments on Dy${}_{2}$Ti${}_{2}$O${}_{7}$ [13]. Of course, the third law of thermodynamics is not actually violated, as the temperature could be lowered further still and the entropy could still vanish. This happens in water ice where the low temperature ice phases have no “proton disorder”. However, at least for temperatures between 0.3 Kelvin and 1 Kelvin, most spin configurations obey the ice rules and the third law appears to be violated.

Ultimately, the thermodynamic measurements of spin ice—its equilibrium properties—agree with theory. The third law is violated as expected at intermediate but low temperatures. Critically, the celebrated discovery of spin ice was questioned in 2013 in a careful study of heat transported through a sample of spin ice during the measurement of the heat capacity [22]. This study claimed the residual entropy vanished, an observation suggesting transitions between ice preserving states were very slow. There must, therefore, be a frozen state down to zero temperature, similar to a glass. However, in 2018, a new study [23] found that if complete equilibrium was established between each heat capacity measurement (by letting 3–4 h pass), the original [13] equilibrium form returned. The authors argue that this occurred because two effects mostly canceled each other out and so transitions between ice rule-obeying states do indeed occur, but for a significant number of them to occur requires multiple hours.

### 2.5. Supercooling and Listening to Monopole Behavior

The fundamentally striking observation that hours are required to equilibrate Dy${}_{2}$Ti${}_{2}$O${}_{7}$ despite a single spin flip process requiring only about 1 microsecond, (ref. [24]) motivated several novel monopole dynamics experiments. If the monopoles exist and are responsible for the transitions between ice preserving states, then they form an electrically neutral fluid. If so, the properties of this fluid are not immediately evident.

To investigate these questions, several Dy${}_{2}$Ti${}_{2}$O${}_{7}$ rings were synthesized [25]. These samples have a single hole through their center, thereby allowing the study of the flow of monopoles around the ring when the magnetization is driven by a superconducting toroidal solenoid—superconducting wires wrapped around one side of the ring as in Figure 4. The researchers [25] were able to show that this flow obeys all the expectations of a supercooled liquid. Glass, warmed up so that it is malleable, behaves exactly the same way.

A second study discovered that the undriven case, a study geared towards measuring the random motion of monopoles as they cross the superconducting loops shown in Figure 4, produces noise at audible frequencies [26]. Converting the data to an audio format enables listening to the motion of monopoles (assuming monopoles are the correct interpretation of the data). This suggests that while monopoles may move from one atom to the next in a microsecond, they take a long time to move over distances of the order of the thickness of the wires in these experiments (0.1 mm) [25,26]. Compared to microsecond time scales of one spin flip, audio frequencies are 1000 times slower, so the noise experiments directly reveal slow monopole behavior.

### 2.6. Spin Ice: A Clean Glass?

A recent study of the magnetic noise using superconducting interference devices (SQUIDs) [27], a simpler but similar setup to the cylinder geometry experiments discussed above, sought to compare the magnetic noise with theoretical expectations to determine if the noise is produced intrinsically by the ice rules or whether other unrelated effects are responsible. In essence, this study pursued the general question: can we have a glass purely from the ice rules, or is Dy${}_{2}$Ti${}_{2}$O${}_{7}$ another glass-like system where random disorder plays a role? They approached this by attempting to explain the data with Monte Carlo simulations of clean disorder-free models. Remarkably, at low temperatures, they found that indeed much of the salient behavior could be attributed to spin flip dynamics, but the comparison, while strong in many respects, was just at the level of the decay rate to equilibrium. This comparison excluded many but not all mechanisms. The researchers concluded, “it is tempting to speculate that here we have, on the one hand, the relatively fast motion of monopoles trapped in their individual surroundings where, however, the confined paths cannot lead to a large change of the magnetization and on the other hand, rare longer-distance excursions of monopoles toward, for example, another local trap” [27]. This raises the question of whether the rare, longer-distance excursions of monopoles are responsible for glass-like behavior. Perhaps the answer can be found in a hybrid approach by coupling exploitation of the kinetic Monte Carlo’s ability to draw samples from the master equation of Equation (Equation 2) (a probability distribution that is hard to evaluate) with dynamic mode decomposition or approximate Bayesian computations, as discussed below in Section 6, Section 7 and Section 9.

## 3. Approximate Hamiltonians for Molecular Dynamics

Material systems present an incredible diversity of complex behavior—ranging from superconductivity and multiferroic properties of transition metal oxides, to high-strength, high-temperature alloys, to catalytic behavior of platinum compounds, to the tunable properties of nanoparticle ionic materials. Designing materials for specific properties necessitates using and understanding these complex interactions. Given the number of chemical species in an alloy—from 2 up to as many as 14 in aerospace alloys—the possible interactions expand as the possible ways of arranging the elements balloon quickly. Arguably, within the enormous phase-space of a material, only a small subset of structures are important for understanding and predicting material properties. The key to representing the complexity in a manageable and understandable form is finding the important structures of a given material. Using this information for material design requires a computationally efficient model for the properties of the representative structures.

Phenomena that reach across multiple scales of length—from microscale to mesoscale to macroscale—and time—from nanoseconds up to years—are a common feature of all material systems. The multiscale nature of materials is a significant challenge to modeling, as no single model is capable of capturing all scales with the same fidelity. Connecting models at different scales is the general problem of “coarse-graining,” where each higher length- or time-scale is modeled using a parametrized, physics-based model. This includes a myriad of problems from atomistic potentials for light-weight alloys, to ionic conductors for energy storage, to the dynamics of proteins in water, to flexible membranes of cells, to the dynamic patterning of dislocations in a deforming metal, to electronic states in a semiconducting device. The challenge of coarse-graining is not in selecting the best set of parameters, but rather in determining the accuracy of a given coarse-graining treatment. Specifically, how the flexibility in parameters for a coarse-grained model reflects itself in the flexibility of possible predictions is unknown as is the extent (if any) of adding new data or expanding a model to improve the predictions of coarse-grained treatment. The special challenge is that often these questions are posed for configurations that are only accessible via the model itself. However, validating a model in the absence of available direct experimental or computational data is highly problematic.

Empirical potentials are “course-grained” models of atomic interactions and are fundamental to materials modeling. They allow molecular dynamics simulations of processes involving 10${}^{6}$–10${}^{9}$ atoms and timescales of nano- to microseconds or longer and are necessary for both length- and time-bridging methods that span orders of magnitude in scale. Their optimization to reproduce computationally demanding quantum-mechanics-based simulation methods is a significantly challenging problem, despite significant advances in finding the best model parameters to reproduce a fit database. For example, algorithms for global minimization techniques (simulated annealing [28,29], genetic optimization [30]), development of flexible models (genetic algorithms [31]), and selection among different models (cross-validation [32]). All of these approaches help ensure that the best model representation is available for a given set of data. These approaches, however, lack the ability to determine whether a model accurately predicts properties outside the optimization database. Recently, the concept of “sloppy models” [33] has proven fruitful for estimating model error [34].

Efforts in material representation have improved the database for optimization, or worked from a data-mining perspective. An early advance in improving the database for model optimization was the concept of force-matching [35], which includes not just energies of defected structures but their forces (derivatives of energy with respect to atom positions). Data-mining approaches use information from other chemically similar systems to make informed guesses for possible structures [36]. This approach is driven by the assembly of a large database of existing knowledge—rather than building knowledge incrementally or using that knowledge to direct the development of more data. More recent work includes approaches for optimization of the fitting database [37] to leverage massively parallel architectures. This algorithm optimizes the target structures and properties, as well as their “weight”, to guide the optimization of a potential to make accurate predictions [38]. This approach has been extended to the development of massively parallel evaluation engines for empirical potentials and been applied in a recent study of empirical potentials [39].

Advances in ML and the automation of the development of large computational databases have driven the field of materials modeling to develop new approaches. In the past, interatomic classical potentials were developed based on physically motivated forms, following from the quantum mechanical treatment of electronic bonding to “integrate out” the electronic degrees of freedom [40,41,42,43,44,45,46,47,48]. While there is a strong benefit in relying on physical intuition in the development of models, the advantages of ML combined with the increasing availability of large-scale computational resources to generate large density-functional theory (DFT)-based databases has greatly changed the approaches to modeling interatomic potentials; see [49,50,51,52,53,54,55,56,57,58,59]. These ML models have, in some cases, achieved nearly DFT-level accuracy at a fraction of the computational cost, but do still require orders of magnitude more computational effort than “traditional” physics-based models to achieve the same accuracy. Moreover, ML models are often largely uninterpretable, acting as essentially data-driven black-boxes. Research continues to both push for more computationally efficient ML models as well as improved interpretability [60,61,62].

ML potentials rely on a general approach to “encoding” the atomic environment around an atom as a “fingerprint”, and then creating a (generally nonlinear) model of the energy for that configuration. The particular challenge of an atomic environment is that the number of “neighbors” of an atom—those within a fixed distance—is variable; moreover, symmetry requires that the energy be independent of the order of neighbors, as well as arbitrary rotations in Cartesian space. Traditional interatomic potentials overcome these difficulties by using prescribed functional forms, such as a sum of functions only of the pairwise distance of atoms (two-body interactions), or including functions of triplets of atoms (three-body interactions). In the case of the various ML methods (e.g., Gaussian Approximation Potentials (GAP) [50], Moment Tensor Potentials (MTP) [55], Neural Network Potentials (NNP) [49], Spectral Neighbor Analysis Potentials (SNAP) [52], and quadratic SNAP (denoted qSNAP) [58]), fingerprints are constructed in more complex ways. For example, NNPs use atom-centered symmetry functions [63], while GAP and SNAP/qSNAP use “smooth overlap of atomic positions” kernels [64] built from a spectral decomposition of spherical harmonics in four-dimensional (4D) space, and moment tensor potentials rely on rotationally covariant tensors [55]. Details of the particular ML interatomic potentials [49,50,52,55,58] and their descriptors [63,64] can be found in the literature.

As an example of a “traditional” classical potential, the spline-MEAM approach—originally developed for elemental Si [28], but has since been applied to Nb, Mo, Ti and Ti-O [29,38,65,66,67]—computes the energy of a configuration of atoms (a set of tuples of positions and chemistries $\left\{\left(R_{i},\alpha_{i}\right)\right\}$) as a sum of pair and embedding terms,
(5)$$E=\frac{1}{2}\sum_{ij}\phi_{\alpha_{i}\alpha_{j}}\left(\right|R_{i}-R_{j}\left|\right)+\sum_{i}U_{\alpha_{i}}\left(n_{i}\right)$$
where the “density” $n_{i}$ at each atom contains both a pair and three-body term,
(6)$$n_{i}=\sum_{j}\rho_{\alpha_{j}}\left(\right|R_{i}-R_{j}\left|\right)+\sum_{jk}f_{\alpha_{j}}\left(\right|R_{i}-R_{j}\left|\right)f_{\alpha_{k}}\left(\right|R_{i}-R_{k}\left|\right)g_{\alpha_{i}}\left(\cos\theta_{jik}\right).$$
Each function ϕ, *U*, ρ, *f* and *g* is a cubic spline with fixed knot points. Therefore, the function values at each knot point become a parameter to optimize. For a single-component system, this produces 5 functions with 5–10 knot points each; for a two-component system, there are 11 functions (3 pair functions) so the number of parameters more than doubles, and the amount of density-functional theory data needed for optimization also increases. This form has shown sufficient flexibility to study both metals and semiconductors, but its use for multicomponent systems remains limited due to the lack of optimized parameters for multicomponent systems. The spline MEAM form itself can be thought of as a superset of other interatomic potentials from Lennard-Jones [40] (LJ), to the embedded-atom method [42] (EAM), the original modified embedded-atom method [43] (MEAM), the famous silicon Stillinger–Weber [45] (SW) potential and Tersoff [44] potentials.

A recent study [39] showed the overlap of the traditional interatomic potentials with ML approaches. An earlier study [68] constructed a Pareto front for ML potentials on a “computational cost" versus "accuracy” plot. This study helped to highlight the tradeoffs in efficiency necessary to maintain high accuracy for some ML approaches. The wide variety of computational effort for the different methods is primarily driven by the complexity of the fingerprints. Traditional interatomic potentials can be considered as providing a simplified fingerprint of an atomic environment, in terms of a few functions evaluated for an atom. For example, the “density” around an atom in spline MEAM, Equation (Equation 6), can be considered an element of a fingerprint of the atomic environment, while the sum of pair energies in Equation (Equation 5) is another fingerprint. Of course, this is an ML viewpoint on the problem of interatomic potentials, which makes a spline MEAM form look rather crude. However, it also indicates that a spline MEAM should be incredibly computationally efficient to evaluate. It was shown that by using large DFT databases intended for training machine learning potentials and applying ML optimization techniques to the spline MEAM form, that accuracy could be dramatically improved for spline MEAM without losing computational efficiency or interpretability. Moreover, the new potentials were themselves on the Pareto front with the ML potentials. This is pushing new research where spline-based fingerprints may be used as the basis for developing accurate and efficient approximate Hamiltonians for molecular dynamic studies.

## 4. Dynamics of Moist Atmosphere

A distinctive feature of the Earth’s atmosphere lies in its active hydrological cycle [69]: water vapor evaporates at the surface of the oceans and is transported upward by atmospheric motion, until it condenses and falls back to the Earth’s surface as rain or snow. Phase transitions account for about 80% of the energy exchange between the Earth’s surface and the atmosphere and are central to many weather phenomena, such as clouds, thunderstorms and hurricanes. Understanding the interactions between atmospheric flows and the hydrological cycle is critical to addressing many fundamental issues in atmospheric and climate sciences, such as determining the climate sensitivity to increase greenhouse gases concentration, predicting the intensity of tropical storms and improving long-term weather forecasting in the tropics.

From a mathematical perspective, the inclusion of phase transitions in the fluid dynamic equations introduces an unusual, non-linear behavior. Indeed, depending on its temperature and content, an air parcel can either be unsaturated—in which case all the water is in gas phase—or saturated, in which case water will be present in a combination of gas, liquid and ice phases. Thus, in contrast with the advection terms that amount to a quadratic non-linearity in the equations of motions, saturation is represented through on–off conditions in the equations of phase. In particular, such non-linearity can lead to scale interactions across a much broader range of scales.

We discuss here two specific problems that show the impacts of phase transition on atmospheric dynamics. We selected highly idealized problems—avoiding some of the complexity resulting from the representations of physical processes such as cloud microphysics and radiative transfer—to focus primarily on the interactions between dynamics and the hydrological cycle. The two problems here focus on two different atmospheric scales: in moist Rayleigh–Bénard, we explore how individual clouds can organize into more weather systems on scales of a few hundred kilometers in the tropics, while in the moist quasi-geostrophic turbulence, we investigate the impacts of phase transition on mid-latitudes weather systems at a scale of 1000 km and larger.

### 4.1. Moist Rayleigh–Bénard Convection

Rayleigh–Bénard convection is a classic problem in physics in which a fluid is heated at a lower, warm boundary and cooled at a colder, upper boundary. Moist Rayleigh–Bénard convection revisits this problem by including an idealized representation for the condensation of water vapor in the air [70]. The inclusion of water vapor leads to a novel convective configuration, known as conditional instability in atmospheric science, in which the layer is stably stratified for unsaturated parcel, but unstable for saturated parcels. The statistical equilibrium state under conditional instability isolates turbulent, cloudy patches surrounded by large quiescent dry regions [71]. More recently, ref. [72] have shown that a rotating version of the moist Rayleigh–Bénard convection spontaneous produces intense vortices, similar to the tropical storms in the Earth’s atmosphere.

The system is the 3D Boussinesq–Navier–Stokes equations with the consideration of phase change of water vapor in a rotating frame constructed in [70]. The equations are given as
(7)$$\begin{array}{ccc} \frac{du}{dt} & = & -\nabla p-fe_{z}\times u+\nu \nabla^{2}u+B\left(D,M,z\right)e_{z}, \end{array}$$
(8)$$\begin{array}{ccc} \nabla \cdot u & = & 0, \end{array}$$
(9)$$\begin{array}{ccc} \frac{dD}{dt} & = & \kappa \nabla^{2}D, \end{array}$$
(10)$$\begin{array}{ccc} \frac{dM}{dt} & = & \kappa \nabla^{2}M. \end{array}$$ Here, $\frac{d}{dt}=\frac{\partial }{\partial t}+u\cdot \nabla$ denotes the material derivative, u is the velocity field, *p* is the kinematic pressure perturbation, *f* is Coriolis parameter, ν is kinematic viscosity and κ is the scalar diffusivity. A dry buoyancy *D* and a moist buoyancy *M* are linear combinations of the total water content and the potential temperature on the unsaturated and saturated side of the phase boundary [70]. The dry buoyancy field *D* is similar to the virtual potential temperature and the moist buoyancy field *M* to the equivalent potential temperature. We apply the no-slip boundary condition for the flow at z=0 and free-slip boundary condition for the flow at z=H. We assign Dirichlet conditions for dry and moist buoyancy with $D_{0},M_{0}$ and $D_{H},M_{H}$ for two buoyancy fields, respectively. The buoyancy field *B* is defined as
(11)$$B\left(D,M,z\right)=\max\left(M,D-N_{s}^{2}z\right),$$
with the fixed Brunt–Väisälä frequency $N_{s}$ that is determined by the moist adiabatic lapse rate. When the layer is unsaturated, the buoyancy takes the dry buoyancy subtracted by Brunt–Väisälä frequency, and the buoyancy takes the moist buoyancy when the layer is saturated. The non-linearity of Equation (Equation 11) captures the discontinuity in the derivative of buoyancy associated with the phase transition [69,70]. The parameter space of this idealized system can be accessed through the six non-dimensional numbers, which includes the moist Rayleigh number ($Ra_{M}=\frac{\left(M_{0}-M_{H}\right)H^{3}}{\nu \kappa }$) and convective Rossby number ($R_{0}=\sqrt{\frac{M_{0}-M_{H}}{f^{2}H}}$). Given the stable stratified, dry buoyancy and the unstable, stratified moist buoyancy, this idealized configuration is capable of producing conditional instability in the Earth’s atmosphere. The solution of equations of motion is obtained through an incompressible numerical solver [73] ([IAMR]) for moderate moist Rayleigh numbers (up to $10^{7}$).

The result reproduces many of the characteristics of moist convection in the Earth’s atmosphere. Unlike the result of classic Rayleigh-Bénard convection that has symmetric updrafts and downdrafts, the moist problem exhibits strong upward motions of moist saturated air parcels in a small portion of the domain, which are compensated for by slow subsidence of mostly unsaturated air parcels over most of the domain. In the absence of rotation, convection aggregates into active patches separated by unsaturated regions (Figure 5a). When rotation is included, the updrafts organize into intense hurricane-like cyclonic vortices surrounded by broad quiescent regions. This regime emerges when the time scale for rotation is about ten times longer than the convective time-scale (Figure 5b). The vortices observed in our simulations exhibit many of the characteristics of tropical cyclones: a warm, moist vortex core with the strongest azimuthal wind near the lower boundary; a strong secondary circulation characterized by low-level inflow; ascent in a circular eyewall; and upper-level outflow. The vortex structure is consistent with theoretical models for tropical cyclones, including the role of slantwise convection in the eyewall [74]. A key finding here is that the emergence of intense vortices in our simulations indicates that tropical cyclogenesis may occur even in the absence of interactions with radiation, surface flux feedback or reevaporation of precipitation, as all of those processes were omitted from our simulation. Rather, our results indicate that the formation and maintenance of hurricane-like vortices involve a combination of rotation and thermodynamic forcing in a conditionally unstable atmosphere.

The exploration of parameter space suggests that to form the hurricane-like vortices requires a sufficiently moist Rayleigh number and marginal rotation, in contrast with rapid rotation in classical Rayleigh–Benard convection. When the rotation is irrelevant, the convection aggregates into patches, and the size of self-aggregated convection increases with the moist Rayleigh number. With the increase in moist Rayleigh number in a fixed domain, the convection turns into intermittent convection. As the rotation becomes relevant, the large patches start to form the hurricane-like vortices associated with secondary circulation characterized by an Ekman flow at the lower boundary.

### 4.2. Moist Geostrophic Turbulence

Much of the atmospheric dynamics of the mid-latitudes arise from the redistribution of both sensible and latent heat by small-scale vorticity dynamics. As these small vortices are propagated by the bulk flow, they elongate and ultimately act to continuously generate planetary-scale features such as the jet stream. Key questions about these systems involve its scales including what mechanisms determine the size of the smallest scale vortices, the speed at which they grow, the maximum size that vortices can attain before being redirected into the jet stream. The answers change depending on the structural features of the atmosphere, including the north–south temperature gradient and the vertical stratification of the atmosphere. Recently, it has been shown that the hydrological cycle can add additional energy to the system, thereby resulting in a more energetic flow with features at both smaller and larger scales than their dry counterparts.

Initially, Lorenz [75] conceived of atmospheric circulation in terms of a cycle wherein the north–south temperature gradient generates motion, which acts to decrease the temperature gradient. Ultimately, the atmosphere settles around a reference temperature gradient based on the balance between vorticity dynamics and the incoming solar radiation. If that gradient becomes steeper than the reference, the atmosphere becomes unstable, with the more energetic vorticity dynamics acting to redistribute heat towards the poles [76]. When the gradient becomes less steep than the reference, the kinetic energy begins to dissipate, reducing the transport of heat towards the poles. Lorenz [77] later updated this concept to include the north–south gradient of humidity. Within this framework, moisture is estimated to account for 30–60% of the energy that gets converted into kinetic energy.

Early intuition about mid-latitude systems came from the quasi-geostrophic (QG) models [78,79,80]. These approaches approximate the atmosphere as (1) a well-stratified system with (2) a north–south temperature gradient and with (3) horizontal pressure gradients that are nearly balanced by the Coriolis parameter representing planetary rotation. Lapeyre and Held [81] explored an updated version of the two-layer QG model that included condensation and latent heat release. The governing equations of the Moist QG model can be expressed in terms of a column-averaged barotropic vorticity $\zeta_{\psi }$ and a vertical gradient baroclinic vorticity $\zeta_{\tau }$. The full system can be expressed as
(12)$$\begin{array}{cc} \frac{D}{Dt}\left(\zeta_{\psi }+\beta y\right) & =-J\left(\tau ,\zeta_{\tau }\right)-R_{\psi } \end{array}$$
(13)$$\begin{array}{cc} \frac{D}{Dt}\zeta_{\tau } & =-J\left(\tau ,\zeta_{\psi }+\beta y\right)-f_{0}\frac{W}{H}-R_{\tau } \end{array}$$
(14)$$\begin{array}{cc} \frac{D}{Dt}\eta & =-W+LP-R \end{array}$$
(15)$$\begin{array}{cc} \frac{D}{Dt}m & =J\left(\tau ,m\right)+W-P+E \end{array}$$ Here, D/Dt represents material advection by the barotropic flow. The streamfunctions ψ and τ correspond to the vorticities as $\zeta_{\psi }=\nabla^{2}\psi$, $\zeta_{\tau }=\nabla^{2}\tau$. $J\left(\cdot ,\cdot \right)$ indicates the Jacobian. The Coriolis parameter *f* is linearly approximated relative to a reference value $f_{0}$, such that $f=f_{0}+\beta y$. Surface dissipation, represented by $R_{i}$, predominantly dampens the system at large scales. *H* is a characteristic height scale of the atmosphere and *W* captures the generation of baroclinic vorticity by vertical motion. Positive values are associated with upward motion, corresponding to convergence of flow near the surface and divergence aloft. The interface η between the top and bottom layers of the model acts as a proxy for the mid-atmosphere temperature, which in turn is the source of baroclinic vorticity. This term can be related to the baroclinic streamfunction as $\eta =2H\lambda^{-2}f_{0}^{-1}\tau$, where λ is the Rossby deformation radius. Diabatic forcings are characterized by the radiative cooling *R*, precipitation *P* and evaporation *E*. The parameter $L\in \left[0,1\right)$ characterizes the relative strength of latent heat release compared with the vertical stratification. When L=1, latent heat release fully compensates for any adiabatic cooling in an ascending parcel, resulting in an atmosphere which is effectively non-stratified.

Precipitation is triggered at locations where the atmospheric moisture content exceeds a saturation value set relative to the temperature by the Clausius–Clayperon relation. This introduces a two-fold non-linearity to the forcings associated with moisture content. We can remove one part of that non-linearity by making a linear approximation of the Clausius–Clayperon, thereby defining the saturation moisture value as
(16)$$m_{s}=C\eta ,$$
where C is the linear constant defining the Clausius–Clayperon relationship here. The precipitation term is then defined as
(17)$$P=\left\{\begin{array}{cc} \frac{m-m_{s}}{\tau } & \mathrm{for}\;m\geq m_{s}\\ 0 & \mathrm{for}\;m<m_{s}, \end{array}\right.$$
where τ is the characteristic relaxation rate of the system, capturing the typical timescale on which a moisture surplus will be eliminated by the precipitation.

In the dry system, typically, only Equations (Equation 12)–(14) are considered. This allows the temperature to act as a source for the baroclinic vorticity. In the full moist system, the precipitation acts as a transfer term between the water vapor content and the temperature. Hence, a complete source of the baroclinic vorticity must include the moisture content. This total source evolves as
(18)$$\frac{D}{Dt}\frac{\left(\eta +Lm\right)}{1-L}=J\left(\tau ,\frac{Lm}{1-L}\right)-W+\frac{LE-R}{1-L}.$$ A special case with $m=m_{s}$ everywhere can be under certain parameter conditions. First, choose R=LE= constant, and choose a very high value for this constant. Second, choose a small value for τ such that the adjustment by precipitation is nearly instantaneous. This corresponds to the Strict Quasi Equilibrium approximation of Emmanuel et al. [82]. Under these conditions, (Equation 18) becomes an evolution equation for η, rescaled by the constant
(19)$$\mu =\frac{1+\mathrm{CL}}{1-L}.$$ This quantity characterizes the vertical stratification of the moisture in an atmosphere saturated everywhere.

A contrast between a dry QG run and a saturated one is shown in Figure 6. The plotted terms are the barotropic vorticity, baroclinic vorticity, and the source is defined in (Equation 18). Compared with that of the dry case, the source term exhibits sharper local gradients in the saturated case, which are converted into baroclinic vorticity. The system becomes more energetic, with more intense and smaller vortices which are driven by the latent heat released from condensation. This also strengthens the turbulent inverse energy cascade, which terminates at a larger scale in the saturated case than in the unsaturated case. We can see this by contrasting the presence of about six organized zonal bands in the dry case (in the upper left and right panels of Figure 6) against against 2 in the saturated case (in the lower left and right panels of Figure 6).

## 5. Urban Engineering with Modern Remote Sensing

### 5.1. Urban Data Acquisition Complexities

Aerial light detection and ranging (LiDAR) is a line-of-sight form of remote sensing, with the resulting point cloud containing three-dimensional (3D) coordinates along with other data and meta-data attributes. Traditionally used for mapping, surveying, and planning purposes in non-urban areas, interest in using LiDAR in urban areas has greatly expanded in recent decades. As part of this (arguably starting from [83]), there has been a growing appreciation for the advantages of multi-pass, aerial flight missions (e.g., [84,85]), as well as combining data from various platforms (e.g., aerial and terrestrial) to overcome platform-based occlusions in disparities in positional data densities. Specifically, as schematically shown in Figure 7, the nadir mounting orientation of LiDAR sensors on aerial platforms introduces an approximate 10:1 bias for the collection of horizontal versus vertical data. Therefore, while modern aerial LiDAR units are capable of collecting in a single pass 30–50 pt/m${}^{2}$ on a horizontal surface (e.g., street, roof), this is likely to translate into only a handful of points on vertical surfaces, far fewer than are needed for traditional object detection algorithms [86,87,88] or for features for input into machine learning algorithms ([89]).

Problematically, the co-registration of multiple flight passes introduces a disproportionate amount of noise into the data set. The authors of [84] refer to this as “cross-pass” error, as opposed to “within-pass” error. The amount is directly influenced by the number of passes being incorporated and distance of the offset distance from the point of interests. Such overlapping as an intentional strategy for more comprehensive vertical, urban data capture is becoming increasingly common (e.g., refs. [90,91,92]).

Smoothing (or denoising) of such data sets is often needed to successfully apply object segmentation and change detection algorithms. Yet, performing such smoothing with traditional algorithms can be problematic as a function of the complexity of urban areas and the absence of an inherent positional relationship between various flight passes (i.e., each point is assigned a position based on positional measurements acquired contemporaneously with only that flight pass).

### 5.2. Machine Learning Prospects for LiDAR

Given the rapidly increasing rate and expanding spatial extent of aerial LiDAR acquisition, especially in the form of open-access, national-level scans [84], there is a strong motivation for investigating the use of ML approaches for LiDAR smoothing, as using least-square methods only obfuscate points that may be in error through what is in effect a hyperlocal averaging or weighting-based fitting, which may ultimately unnecessarily introduce errors that then propagate through sections of the dataset, if followed by larger geometric fitting (e.g., ground plane determination). While machine learning approaches hold an inherent appeal, problematically, the most fundamental issues about how to handle input training data have yet to be addressed, thus hindering effective adoption of ML techniques. Specifically, most research to date related to ML and LiDAR has been limited to the straightforward applications of a handful of well-established techniques, such as random forest, support vector machine and k-means clustering, to relatively small and fairly homogeneous datasets without engaging in the fundamental questions related to (1) the nature of the input data (volume, nature and dimensionality), (2) the needs for transferable and adaptable approaches to reflect the complexity of built environments; and (3) the return on investment aspects with respect to both data preparation and processing. Instead, they are generally interested in improving the F1 score for only a narrow set of object classes within one or two data sets.

This relatively shallow engagement with ML/DL learning is problematic in that communities that wish to benefit from ML/DL techniques receive little guidance in rigorous method selection and optimized implementation. These factors can lead to the adoption of less than ideal techniques and the expenditure of unnecessary resources labeling and processing data only to incrementally improve accuracy but still failing to gain the necessary insights into developing ML/DL-based strategies that can be used robustly and for real-world applications and data sets, instead of “toy” problems tested within highly controlled and limited conditions, where significant manual parameter tuning is needed to obtain satisfactory outcomes.

While the application of ML to imagery is well established, fairly thoroughly understood, and widely adopted, aerial LiDAR capture introduces fundamental complexities that preclude a rigorous, consistent selection of techniques and input parameters (especially related to the characteristics and size of the training data sets) and potential reusability of trained ML models to multiple LiDAR data sets (even those documenting the same spatial extent). These challenges can be roughly categorized as stemming from six aspects: (1) data acquisition decisions, (2) data quality issues; (3) training data characteristics; (4) heterogeneous nature of urban areas; (5) data access; and (6) data processing challenges.

To date, ML has predominantly been applied to text and imagery, where dimensionality is low, the samples are readily available, and the metadata are provided in a complete and consistent manner. While efforts to apply ML to LiDAR and other three-dimensional (3D) remote sensing data have shown promise, not only have publications to date not addressed many of the fundamental questions including the required size and composition of training data, but the LiDAR data’s high dimensionality, discontinuous nature, heterogeneity of distribution, and potential spatial incompleteness in 3D (based on acquisition platform), are characteristics that differ fundamentally from text and imagery. The problem is complicated by the relative paucity of labeled data sets, especially when working at scale (e.g., in an urban environment). Thus, while ML has been shown to be potentially more effective than traditional algorithm development, a rigorous exploration of the topic has yet to be undertaken, and efforts in this direction have generally been limited to a single aspect of this laundry list of challenges

For urban datasets, there are inherent challenges to developing training sets. As mentioned previously, there is a major data capture imbalance. This means that the amount and quality of representation of certain objects is inherently better than others (e.g., streets versus facades). Furthermore, urban areas are inherently heterogeneous. This is true of both their overall form (e.g., the percentage of vegetation in an urban area is only a small fraction of that comprised of roads and buildings) and the composition within the various classes (e.g., skyscrapers versus single family homes). Additionally, training on built environment objects is complicated by the diversity of materials and architectural forms used worldwide.

Even without such complexities, the use of ML for LiDAR is relatively superficially understood to date with respect to the size of training sets, the number of classes, the issue of class balancing, the recommended number of points per object, the uniformity of those points, the segregation of objects and the used of hand-crafted features, as noted by [93].

Some attempts to address these issues use statistically calculated features of the point cloud (e.g., volume, density and eigenvalues in three directions of each object) to which a neural network is then applied. However, crafting features by hand is laborious and not suitable for very large, dense data sets; current aerial data sets can exceed 1 billion points per square kilometer. In addition, hand-crafted features consider points in isolation without ensuring that there is consistency with neighboring points, which can result in noisy and inconsistent labeling [94]. For instance, within an object with points mostly classified as tree, there may be several “noisy” points that should be classified as a person.

The authors of [95] developed a neural network that directly acts on point clouds rather than converting data into 3D voxels or relying on hand-crafted statistical features. This accelerated processing but has to date provided only minimal insight as to optimal training approaches, with most researchers limiting themselves to recycling previously labeled datasets due to the high cost of labeling. This comes more than a decade after [96] established that in the comparative analysis of machine learning models, results that are reported on a fixed-size training data set do not provide any information on how the model would fare with differing sizes of training data. PointNet-derived neural networks also report results in this way—using the same data set and training size. The insight on varying training data size is important to ascertain the reliability of the model in new applications, where the amount of training data available is not always the same as that on which the model was trained. The performance of a machine learning algorithm can be quantified by a learning curve, which benchmarks a generalization performance metric, such as accuracy or error, against the quantity of training data [97]. By illustrating the effect of training different amounts of data, one can determine at what points the amount of training data is considered sufficient, redundant, or causes overfitting. The absence of a theoretical foundation regarding learning curves has been investigated to some extent for neural networks in general. The authors of [97] found that neural networks are often built for a specific application, focusing solely on performance, and achieving this higher performance by modifying the parameters based on trial and error, rather than on theory.

In conclusion, to date, little effort has been spent on evaluating the characteristics of the training of ML approaches. As such, many fundamental questions on how performance is impacted by changes in the quantity of input data remain unexplored. An implementation’s performance should also be further thoroughly examined by using uneven distributions across object categories, density of points, and number of points per object, which more closely resembles remote sensing data. Thus, the problem is not necessarily in the ML approach but in the blind application of existing implementations. The author’s experience to date has raised critical questions as to the inherent feasibility of transfer learning, at least as how it is presently practiced in the imagery community.

## 6. Spatiotemporal Prediction Techniques

When considering the issues of spatiotemporal prediction, it is important to realize all of the differing options currently available. Most of the methods listed in this section have existed in the community for over 70 years, however, the problems being addressed today are of much greater complexity (dimension) and are being driven by the exponentially increasing data acquired at multiple resolutions and time scales [98]. Further, how systems are modeled is changing rapidly as limitations from numerical techniques may be addressed by the vast amounts of data now available. The question today becomes “How much data are required to meet the appropriate prediction window needed?” As such, a review of these techniques may reveal new combinations or *hybrid* schemes for better prediction. Presented here is a brief overview [99]:1.Traditional particle evolution schemes—Classical Mechanics [100,101]:(a)Particles systems defined by either a two-body force, $F({\Delta r}_{\mathrm{jk}})$, or a potential energy, $V(r_{k}),$ given $r_{0},v_{0}$:i.$r_{k}\left(t\right)=\frac{dv_{k}}{dt}$ii.$v_{k}\left(t\right)=\frac{da_{k}}{dt}$iii.$a_{k}\left(t\right)\to \mathrm{via}\;\;\sum_{j}\;F\left({\Delta r}_{jk}\right)\to \mathrm{via}\;\;V\left(r_{k}\right)\to (\mathrm{back}\;\mathrm{to}\;i.)\;(\mathrm{repeat});$(b)Dynamical Systems [102,103]:$\dot{r}=f\left(r,t\right)$;(c)Euler–Lagrange equations of motion;(d)Hamiltonian formulation.2.Field evolution systems u(r,t) [104]:(a)Partial Differential Equations (PDEs):      $\frac{\partial u}{\partial t}=f\left(u,r,t\right);$(b)Finite Difference/Element Equations:      u(t+Δt)=g(u,r,t).3.Forward Time Evolution Operator—$\hat{O}\left[\Psi \left(t\right)\right]$ [105,106](a)via Linear Operations (Markovian);(b)via Gaussian Processes (GP);(c)via Machine Learning (ML);(d)via Discrete Operations (i.e., game of “Life” by Conway) [107,108].4.Modal Decomposition—numerous applicable techniques for time-dependent amplitudes/power spectra.(a)Traditional analytic modes—Fourier, Bessel, Spherical Harmonics, Laguerre, Legendre;(b)Generalized modes—Galerkin methods;(c)Data/Experimentally Constrained Modes—Empirical Orthogonal Functions (EOF, POD, PCA, etc.);(d)In each case, the modes are spatial and static while the Amplitudes, $A_{n}\left(t\right)$, calculated are time-dependent;(e)When constrained by physical principles, the modes must evolve smoothly—no discontinuities.5.Virial Theorem—Used to preserve the statistical features of a system’s time evolution(a)Couples the time-average of the kinetic energy of a system of particles to its internal forces;(b)For systems in equilibrium, relates the kinetic energy to the temperature of the system;(c)For power law-based forces, relates the kinetic energy to the potential energy;(d)Used in systems where the total energy is difficult to account for, but the time-averaged system is well known (i.e., stellar dynamics, thermodynamics).6.Dynamic Mode Decomposition (DMD) [109,110]—Essentially a combination of a Time Evolution Operator applied such that the result is both data-driven and modal—with time-*independent* amplitudes, $A_{n}$, (yet driven in time exponentially). Similar in technique to EOF.7.Physics-Informed Neural Networks (PINN) [111]—Related to spatiotemporal prediction schemes by directly extracting the components of a partial differential equation selected from a library of differential operators. Assists in understanding how a system is evolved over time by determining the equation of evolution.8.Quantum Mechanics [112]—an admixture of a complex Field evolution via a PDE(Schroedinger/Dirac equation) with a statistical interpretation.9.Hybrid Approaches(a)Given the methodology (toolset) listed above, find a suitable architecture to apply that yields the best interpretation of the system’s evolution;(b)As an example, consider a data-driven approach, where a Hopfield NN is employed to reduce the complexity of a large time-dependent data set. Assuming the data follow a particular model, by targeting a particular reduction in order, first project the current state vector of the data onto the best orthogonal domain appropriate to the problem—usually achieved by a linear operation (LA). Next, map the data from an input space to some desired output space using either GP or ML, where the output space contains far less information (Model Order Reduction, POD, Krylov subspace reduction). This step is effectively Variational Auto-Encoding (VAE). From this mid-point state vector, begin to unpack the information by performing the inverse functional mapping ($GP^{-1}$ or $ML^{-1}$) finally bringing the state vector back to the linear space and via a linear inversion, back to the original input data space—effectively decoding the state from the VAE.(c) 
$$\underset{output\;=\;input}{\underbrace{\begin{array}{c} (\;\begin{array}{c} O \end{array}) \end{array}}}\;\begin{array}{c} =\;{\hat{LA}}^{-1} \end{array}\left[{\hat{GP}}^{-1}\left[\begin{array}{c} \underset{\begin{array}{c} midpoint\\ state \end{array}}{\underbrace{\left(\;\begin{array}{c} V\\ A\\ E \end{array}\right)}}= \end{array}\;\underset{\begin{array}{c} Functional\\ Mapping \end{array}}{\underbrace{\begin{array}{c} \hat{GP} \end{array}}}\underset{LinearOp}{\underbrace{\begin{array}{c} \hat{LA} \end{array}}}\underset{input}{\underbrace{\begin{array}{c} (\begin{array}{c} I \end{array}) \end{array}}}\right]\right]\;$$ Through this process, the spatial and temporal features are mapped onto a new subspace (similar to a Fourier transform), where the prediction mechanism could be simpler in its handling of the numerical techniques, as well as error propagation by using properties of the transformed space to track/restrict errors. Currently, we enjoy the computational power to attempt methods such as this.

Completing this survey of modern approaches, consider the understanding gained *during* the process of optimizing these techniques. Typically, a functional mapping is obtained through a feedback loop mechanism, whereby a cost measure is optimized while parameters are adjusted to the mapping. In the case of a NN, the weights and bias vectors are altered via a back-propagation scheme; for Gaussian processes, the covariance matrix is determined; for data assimilation, targeted values within a model are adjusted, etc. During the optimization phase, should the system not converge, appropriate reduction in the functional mappings’ complexity (whether GP, NN-based or perhaps Kalman-filtered) addresses the issue of determining exactly in which space the data manifold truly operates. Choosing too high of a dimensional representation for the functional mapping (GP/ML) introduces unwanted noise into the system, preventing convergence. Choosing too low of a dimension prevents the system from reaching the optimal cost function, as necessary features are being omitted.

Table 1 is an attempt to outline the features of each of the schemes discussed, as well as how the results from each approach compare to one another. From the point of view of spatiotemporal prediction, classifying features falls broadly into two categories: features that determine *how* the method is employed and features of the *results* from those methods.

In the first category, classification is based on the conditions that the methodology adheres to:1.Linear or Non-linear-based;2.Data-Driven or Equation-based (or both);3.Deterministic or Statistical—as a process;4.Continuous or Discrete—numerically and/or analytically;5.Boundary Conditions—fixed or dynamic;6.Constraints—strong/weak/statistical in nature.

Classifying the analysis and results is the second group:1.Analysis type—Least squares (fitted) versus Projection (onto a basis set);2.Predictive—yes or no—this is in contrast with results that are more probabilistic in nature (those that generate a PDF/ensemble result;3.Is a PDF generated?;4.Simulation—versus a direct future prediction from a data-driven result (time-series/forecasting);5.Interpolation—is the method good or bad at interpolation?;6.Extrapolation—how well does it perform (good/bad)?;7.Quality of result:(a)Is the technique excellent at forecasting over a short time-frame?;(b)Forecasting over a long time-frame?;(c)Ensemble forecasting—out of 100 possible simulated projections, x% show the following…;(d)Is some form of feedback used from data (Kalman, Data Assimilation) [115,116,117].

To be complete, a large catalog of features are missing, but in the broadest sense, this attempt is to classify each of the techniques listed in (Table 1). Further, this shows a generalization of the techniques by classes of approaches; however, a second table is added to map some specific techniques that are hybrids of the generalized classes (Table 2). Please note that the breadth of these two tables cannot fully be covered by the collective expertise of the authors listed. As such, these tables represent a “best guess” at classification and are presented here to inspire a conversation within the community on how to best group and classify the state of affairs in predictive methodologies. The hope is that the comparison amongst techniques will inspire and cross-pollinate new approaches.

## 7. Dimension Reduction Techniques

Modern datasets, such as those generated by high-resolution climate models or space observation, can include more than a billion independent variables. In addition to the technical difficulties resulting from dealing with such a massive volume of data, discovering the mathematical and physical relationships that underlie high-dimensional datasets presents a tremendous intellectual challenge. We will review here a set of highly successful approaches aimed at identifying lower-dimensional subspaces that can be used to infer key features of the datasets.

We consider here a dataset of the form
$$X=\left[x_{0},x_{1}\ldots x_{m}\right]$$
composed of *M* vectors $x_{t}$ in $R^{N}$. We have in mind here a sequence of *m* chronological snapshots of observations spread over *N* spatial locations and aim at discovering preferred patterns of spatiotemporal fluctuations. **Principal Component Analysis** (PCA) is probably one of the most commonly used techniques to extract such patterns. In effect, one can view the dataset X as a N×M matrix, and perform a Singular Value Decomposition to rewrite it as
(20)$$X=U\Sigma V^{T}$$
with $U\in R^{N\times r}$, $V\in R^{M\times r}$, and $\Sigma \in R^{r\times r}$, and *r* is the rank of the matrix X. The columns of U are orthogonal and are referred to as the Principal Components (PCs). They can be interpreted as preferred spatial patterns of variations of the datasets. The columns of V are also orthogonal and contain the temporal evolution of each principal component (PC). The matrix Σ is diagonal, and its elements (the singular values of the matrix X) correspond to the square root of the variance associated with each principal component.

The application of the Singular Value Decomposition (Equation 20) recasts the data in a new orthogonal basis made of the principal components, but does not by itself reduce the size of the dataset. This can be achieved by keeping only a smaller number k<<r of principal components associated with the largest singular value. Due to the orthogonality of the PCs, the process is straightforward to project the original data onto a smaller number of PCs that can still be associated with most of the original variance, while excluding directions that are associated with only small fluctuations.

While PCA provides a practical way to identify the main patterns of variability in a dataset, it does not tie any of these patterns to specific dynamical features. One could do so by exploiting the fact that the dataset arises from a time series and assuming that each sample can be generated from the previous one through an unspecified function or set of functions, i.e.,
(21)$$x_{k+1}=f\left(x_{k}\right)$$
and use the data to identify *f*. To make the problem tractable, one must however restrict the set of possible functions. In dynamic mode decomposition [109,136], one assumes *f* to be linear:(22)$$x_{k+1}={\mathrm{Ax}}_{k},$$
where A is a N×N matrix. To determine the matrix A, one can first rewrite the dataset as two large N×(M−1) matrices
$$X_{0}=\left[x_{0},x_{1}\ldots x_{m-1}\right]\mathrm{and}X_{1}=\left[x_{1},x_{2},\ldots x_{m}\right]$$
so that the dynamical relationship (Equation 22) becomes
(23)$$X_{1}={\mathrm{AX}}_{0}.$$ The data matrix $X_{0}$ may not be invertible, but one can nevertheless solve for the matrix $\tilde{A}$ that minimizes by multiplying (Equation 23) on the right by $X_{0}^{†}$, the pseudo inverse of $X_{0}$:(24)$$\tilde{A}=X_{1}X_{0}^{†}=X_{1}X_{0}^{T}{\left(X_{0}X_{0}^{T}\right)}^{-1}$$ This approach is known as linear inverse modeling. One difficulty lies in that the matrix $X_{0}X_{0}^{T}$ is often ill-conditioned, making the computation numerically difficult. To circumvent this problem, one can perform a singular value decomposition on $X_{0}=U\Sigma V^{T}$, then restrict the computation to its r largest singular values as to approximate $X_{0}$ as
$$X_{0}\approx U_{r}\Sigma_{r}V_{r}^{T}$$ We can then approximate the forward matrix as
(25)$$\tilde{A}\approx X_{1}V_{r}\Sigma^{1-}U_{r}^{T}$$ Alternatively, it is often convenient to work directly in the principal component space, which amounts to rewriting the matrix as
(26)$$U_{r}^{T}\tilde{A}U_{r}\approx S=U_{r}^{T}X_{1}V_{r}\Sigma^{-1}$$ Once the forward operator has been estimated, either as S or $\tilde{A}$, then one can then determine its eigenvalues $\lambda_{i}$ and corresponding eigenvectors $v_{i}$
(27)$${\mathrm{Sv}}_{i}=\lambda_{u}v_{i}$$ As $\tilde{A}$ are related through an orthogonal transformation, they have the same eigenvalues, and the corresponding eigenvectors of $\tilde{A}$ are given by $U_{r}v_{i}$. These eigenvectors and eigenvalues are the preferred temporal evolution patterns, as diagnosed through the lag-correlation within the datasets and are referred to as *dynamical modes*.

The eigenvalues offer a useful way to categorize the dynamical mode. When $|\lambda_{i}|\geq 1$, the corresponding mode is growing exponentially over time, which can arise for instance if the datasets have been generated from simulations of hydrodynamical instability. For datasets that are statistically steady, the eigenvalues should all have a norm of less than one, with larger eigenvalues corresponding to more persistent patterns of variations. Complex eigenvalues are associated with oscillatory fluctuations, with a frequency that can be assessed from the angle of the eigenvalues. These complex eigenvalues are associated with complex eigenvectors whose real and imaginary components determine the hyperplane in which the oscillations are taking place.

The dynamical modes can be used to reduce the dimensionality of the datasets, while preserving some of its key dynamical features. In effect, one identifies a number *s* of dynamical modes associated with the largest, absolute eigenvalues and projects the original datasets in the subspace spanned by these eigenvectors. If $W=\{v_{0},v_{1},\ldots ,v_{s}\}$ is the matrix composed of these eigenvectors, then
$$z_{k}=W^{†}U_{r}^{T}x_{k}.$$

## 8. Multi-Resolution Gaussian Processes

Fields such as physics have often relied on decades of theoretical development to reap the benefits of mathematical equations that define physical phenomena. Many physical science domains rely on process-based models or statistical models for data analysis and deriving inference about the unknown components. Physical models are often implemented using numerical simulations that focus on approximating the behavior of the analytical physical equations, which often incur heavy computational costs. To circumvent the need for extensive theory development and expensive resource requirements, utilizing heterogeneous sources of data can enable better generalization capabilities and robustness in modern ML methods. These heterogeneous sources of datasets can make use of any cross-associations among the variables.

Real-world data may exist at multiple spatial and temporal scales. These multiple scales of data may act as a source of heterogeneous information [137]. Very fine resolution data may afford better insights, but may be difficult to obtain. At the same time, coarse-scale data may be easier to obtain, but may provide inexact insights. Therefore, joint, efficient modeling of heterogeneous types of data could be advantageous in an attempt to reap the benefits of multi-scale datasets.

Many existing physical systems are governed by interdependent physical processes. These may be specified as coupled differential equations that may require employing simulation methods to study the evolution of the physical system at different scales and resolutions. Depending on any resource and time constraints, physical system evolution can alternatively be modeled using data-driven machine learning methods.

More modern deep learning architectures employed in the physical sciences are enabled to be designed not just to act as black-box algorithms, but to obtain estimates and predictions that are consistent with known scientific facts and pertain to reality. This requires constraining the ML techniques to produce results that are compatible to the real world. These methodological choices can either be in the form of transformations and feature selection to address observational biases, architectural choices (inductive biases—based on physical laws, boundary conditions) or other implicit learning biases (careful architectural choices to indirectly enforce physical constraints) [138].

An example of learning in the context of noisy and potentially biased observational data is in the use of the encoder–decoder framework for multiscale, micro-climate prediction [139]. The framework utilizes sensor-based, fine-scale (fine spatial and temporal resolution) weather and soil moisture sensor data along with coarse-scale (coarse spatial resolution) local weather station readings to forecast weather at very fine temporal resolutions. Coarse weather station data present higher biases due to their distance from the location of interest (agricultural farms), while the fine-scale sensor data are scarce due to higher costs.

Similarly, incorporating prior domain knowledge or physical constraints in model objective to improve neural network performance is an example of an inductive bias technique, one that has been extensively used [140,141,142,143,144,145,146,147,148]. As an example of multi-fidelity systems, consider a composite neural network framework [146] that learns each fidelity using one neural network and also learns the cross-correlations among the fidelities using a neural network. Another neural network is then used to encode any partial differential equations (PDE) of interest for estimating unknown parameters in the PDE. These models also have something in common with surrogate modeling—several physics-informed machine learning paradigms approximate the input–output relation while minimizing any data needs, especially in the case of high-dimensional, stochastic inputs [142,144].

Graph neural networks [149,150] can also be extremely useful for learning interaction among multi-scale physical processes. They may also offer data assimilation mechanisms to augment the sparse, fine-scale datasets [151]. Probabilistic graphical models [145,152] can also be utilized to learn any prior knowledge about the variables. Additionally, graph neural networks can be used for operator learning in physical problems. These neural operators can learn an output mapping, despite the initial input space discretization. This makes neural networks more flexible than conventional PDE solvers. A neural operator can, therefore, be learned on one discretization and then evaluated on another discretization of the input domain. The trade-off can be perceived in the training time complexity. These also find usage in multiscale modeling [153,154].

As a Bayesian, non-parametric approach [155], Gaussian processes (GP) enable uncertainty estimation. Due to its analytical tractability and flexibility in terms of the kernel and prior choice, GPs enable a principled measuring of similarity via the covariance matrix. This has also led to their applicability in modeling partial differential equations [156,157,158,159,160]. In a multivariate GP setup, a conditional functional posterior estimation can allow for joint modeling of multi-scale datasets. ML may be used for both learning the PDEs from noisy data and modeling the uncertainty propagation through such algorithms using multi-resolutions GPs and modern dimension reduction techniques [161,162,163,164,165,166,167,168,169,170].

To model strongly coupled physical processes, linear multi-output, multi-variate Gaussian process models have been used [137]. There, the higher-fidelity process is modeled as a sum of two components: one that captures its relation with the lower fidelity process and another that is independent of it. Mathematically, the higher-fidelity model (H(x)) can be formulated as,
(28)$$H\left(x\right)=\rho \;L\left(x\right)+\epsilon_{H}\left(x\right).$$ Here, ρ is the linear correlation between high- and low-fidelity processes, L(x) is the GP modeling the output of the low fidelity process and ϵ(x) is a GP that is independent from L(x). This auto-regressive scheme [137] allows for building an *intrinsic corregionalization model*. Therefore, we can jointly model two consecutive fidelity processes,
(29)$$\left(\begin{array}{c} L(x)\\ H(x) \end{array}\right)\;∼\;GP\left(\;\left(\begin{array}{c} 0\\ 0 \end{array}\right),\;\left(\begin{array}{cc} K_{L} & \rho K_{L}\\ \rho K_{L} & \rho^{2}K_{L}+K_{\epsilon } \end{array}\right)\;\right).$$ As shown in (Equation 29), the assumption of linear correlation between two levels of processes aids in leveraging the multivariate GP for enhanced learning of a sparsely available, high-fidelity (or finer-scale) process.

Non-linear coregionalization has also been used to model multi-fidelity, high-order GPs [171]. Following the introduction of multi-fidelity GP frameworks [161,162,167,172], this methodology was modified for multi-output problems. In such configurations, the non-linear association among different fidelities is captured using a deep matrix GP prior. Subsequent fidelities are learned as non-linear, functional approximations of lower fidelities, and the biases are propagated through the fidelities. Unlike this method, the nonlinear autoregressive multi-fidelity GP regression (denoted by NARGP by [161]) was formulated as a group of disjointed GPs (instead of one deep GP) for modeling different fidelities. Based on the auto-regressive scheme [137], the higher-fidelity physical processes employ the mean response estimates from lower fidelity datasets and joint covariant estimates to learn the predictive mean and variance at higher levels.

However, note that GP models scale poorly with data size, and are often cumbersome to use for this reason. Some modeling strategies have been suggested to address this issue: these include coupling GPs with dimension reduction techniques, or using sparse covariance, sparse Cholesky Factorization, sparse precision and low rank approximations of different kinds [155,173,174]. Of particular interest are approaches that leverage the inverse covariance (precision) matrix, since these models retain the physical interpretability and simplicity of GPs, they automatically obtain a partial correlation-based network structure that in many situations may be interpreted to be a causal graphical model, and they are computationally fast. Examples of such models motivated by statistical physics, short-range coupling Hamiltonians (e.g., the Ising model) and ones which offer significant computational advantages may be found in [175,176].

## 9. Approximate Bayesian Computations

The Bayesian paradigm is often the most convenient and appropriate for spatiotemporal modeling. However, Bayesian modeling and the related optimization and uncertainty quantification procedures typically involve computations that are both voluminous and complex. Generally, in the Bayesian framework, all unknown quantities are associated with a probability distribution. This often creates a complex hierarchy of probability distributions. Numerical approximations to conditional probability measures are then obtained using Markov Chain Monte Carlo (MCMC) techniques, which generate a Markov stochastic process on the set of all unknown quantities, with the condition that the stationary distribution of the Markov process is the joint probability distribution of interest. Convergence to the stationary distribution may be challenging, both from a Markov process theoretical viewpoint, as well as in terms of practical guidelines for algorithmic explorations.

Approximate Bayesian computations (ABC) provide an alternative to MCMC computations. As the name suggests, they offer only approximations of the probability distributions of interest in Bayesian analysis. On the other hand, the computations are often much simpler than MCMC alternatives. In this section, we present an exposition on Bayesian techniques and ABC.

### 9.1. Bayesian Analysis

The Bayesian philosophy for modeling typically assumes that the distributions $P_{\theta }$ governing *Y* admit likelihoods p(·∣θ) conditional on parameter Θ∈H (e.g., $R^{p}$), which is itself considered a random variable. We assign to Θ a prior distribution Π(·) with density π(·), thereby obtaining the following hierarchical model:(30)Θ∼π(·),(Y∣Θ=θ)∼p(·∣θ). Inference made on Θ, after observing *n* datum $y_{0}=\left[y_{1},\cdots ,y_{n}\right]$, is encoded in its posterior distribution ${\bar{\Theta }}_{n}∼\Pi \left(\cdot |y_{0}\right)$. Its associated density is given in terms of the likelihood and the prior distribution by Bayes’ Theorem:(31)$$\pi \left(\theta |y_{0}\right)=\frac{p\left(y_{0}|\theta \right)\pi \left(\theta \right)}{m(y_{0})}$$
where m(·) is the marginal distribution of *Y*. The choice of prior is a critical matter for Bayesian techniques. However, a wealth of principles exist, as well as accompanying criticisms exist and guide this choice (see [177]).

A rich set of sampling methods has been developed to broaden the scope of models for which meaningful posterior inference is possible. When (Equation 31) is not analytically tractable, (1) importance sampling, (2) rejection and acceptance sampling and (3) Markov Chain Monte Carlo samplers can fill the gap. Typically, this only requires that the likelihood $p(y_{0}|\theta^{\prime })$ be calculable for the proposed values $\theta^{\prime }$ (more can be found in chapters 10, 11 and 12 of [178]). As an example, Algorithm 1 introduces a basic rejection sampler, which is modified in Algorithm 2 for a likelihood-free Bayesian posterior inference. Suppose the goal is to sample values of *X* following distribution *f*, but this cannot be achieved directly. However, suppose sampling X∼g and *g* is such that f(x)>0⇒g(x)>0 is relatively easy. Then, Algorithm 1 will produce samples from *f*. Further details and extensions of this algorithm can be found in [179,180] and elsewhere.

**Algorithm 1:** Accept/Reject Sampler.

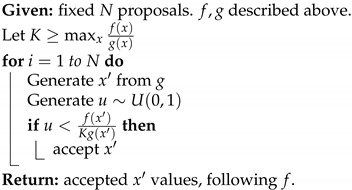



**Algorithm 2:** Likelihood-Free Posterior Sampler ([180]).

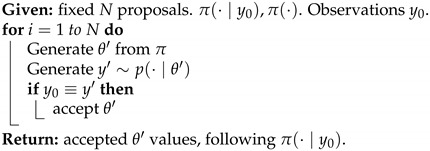



Suppose that for each proposed $\theta^{\prime }$, surrogate data can be generated according to the likelihood $y^{\prime }∼p\left(\cdot |\theta^{\prime }\right)$. This may be possible even for models arising from complex physical phenomena. Then, notice that accepting $\theta^{\prime }$ in Algorithm 1 with the probability proportional to $p(y_{0}|\theta^{\prime })$ is equivalent to accepting $\theta^{\prime }$, if $y_{0}\equiv y^{\prime }$. Algorithm 2 implements this and produces samples from the desired posterior.

This motivates the relaxation, thereby imparting Approximate Bayesian Computation its name. The equivalence condition $y_{0}\equiv y^{\prime }$ is permissible, if we are able to generate synthetic data $\left(\theta^{\prime },y^{\prime }\right)$, but cannot explicitly calculate $p(y^{\prime }|\theta^{\prime })$, as Algorithm 1 requires. Of course, accepting and rejecting vis-à-vis $y_{0}\equiv y^{\prime }$ is only practical for discrete distributions where equality occurs with positive probability. For continuous distributions, $P(Y^{\prime }=Y|\theta^{\prime })=0$, we must relax the acceptance criterion to $y_{0}\approx y^{\prime }$ in some manner. Thus is born the ABC approach and the myriad means to implement and measure this approximation.

Section 9.2 introduces popular algorithms for conducting Approximate Bayesian Computation. As a guiding framework, consider the following intuitive understanding of ABC: accepting simulated $\theta^{\prime }$ only when $y_{0}\equiv y^{\prime }$ exactly returns a sample from the desired posterior. Alternatively, accepting every $\theta^{\prime }$, no matter how closely $y^{\prime }$ resembles $y_{0}$, merely generates samples from the prior. In practice, ABC samplers return $\theta^{\prime }$ such that $y^{\prime }$ is “close” to $y_{0}$ in some sense. The field is currently engaged in characterizing the nature of such a compromise and determining under what conditions these methods be used for meaningful posterior inference.

### 9.2. Different Implementations of ABC

#### 9.2.1. Fixed Cutoff Sampler

The equivalence condition $y_{0}\equiv y^{\prime }$ governing acceptance in Algorithm 2 is impractical for many discrete distributions and impossible to require for continuous data. Algorithm 3 (from [181]) relaxes this condition in a straightforward way. Let $S:Y^{n}\to R^{d}$ be a statistic of the data (sufficient or otherwise) and $\rho_{n}$ a distance metric on its range. Instead of strict equality, Algorithm 3 accepts $y^{\prime }$ if $\rho_{n}\left(S\left(y_{0}\right),S\left(y^{\prime }\right)\right)<\varepsilon$ for a prespecified tolerance value ε, which governs the compromise made to make approximate posterior inference described above. Algorith 3 produces samples $\left(\theta^{\prime },y^{\prime }\right)$ from the joint density ([180])
(32)$$p\left(\theta ,y|S\left(y_{0}\right)\right)\propto \pi \left(\theta \right)p\left(y|\theta \right)1_{\left[\rho_{n}\left(S\left(y_{0}\right),S\left(y\right)\right)\leq \varepsilon \right]}.$$

Typically, we are only interested in samples $\theta^{\prime }$ and discard the accepted $y^{\prime }$. In this case, we have samples $\theta^{\prime }$ from the ABC approximation to the posterior, as given below.
(33)$$\begin{array}{cc} \pi_{ABC}\left(\theta |s_{0}\right) & :=\frac{\pi \left(\theta \right)\int_{B_{s_{0}}}p\left(s|\theta \right)ds}{\int_{R^{p}}\pi \left(\theta \right)\left[\int_{B_{s_{0}}}p\left(s|\theta \right)ds\right]d\theta } \end{array}$$
where $p\left(s|\theta \right):=\int_{Y^{n}}\delta_{s}\left(S\left(y\right)\right)p\left(y|\theta \right)dy$ and $B_{s_{0}}:=\left\{s:\;\rho_{n}\left(s_{0},s\right)<\varepsilon \right\}$.

Note that if *S* is sufficient and ε≡0, then $B\left(s_{0}\right)=\left\{y_{0}\right\}$ and (Equation 33) reduces to the desired posterior exactly.

**Algorithm 3:** Summary Statistic Rejection Sampler (Algorithm D in [181]).

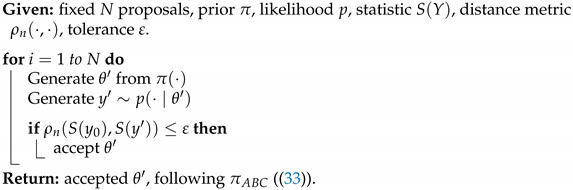



#### 9.2.2. ABC as K-Nearest Neighbors

In practice, the tolerance ε above is often chosen post-hoc, as a percentile of the generated samples. For example, the R implementation abc.abc()[182] by default returns the $\theta^{\prime }$ such that $S(y^{\prime })$ are amongst the $k_{N}=\left\lfloor N/10\right\rfloor$ nearest neighbors of $S(y_{0})$.

The kNN flavor of Algorithm 3 creates the same samples $\left(\theta_{1},y_{1}\right),\ldots ,\left(\theta_{N},y_{N}\right)∼\pi \left(\cdot \right)p\left(\cdot |\theta \right)$. However, it accepts $\theta_{\left(1\right)},\ldots ,\theta_{\left(k_{N}\right)}$ corresponding to the $k_{N}$ nearest neighbors $S_{\left(1\right)},\ldots S_{\left(k_{N}\right)}$ of $S(y_{0})$.

The resulting distribution governing the accepted samples (analogous to (Equation 33)) is given by [183]. Let $d_{\left(k_{N}+1\right)}$ be the $\left(k_{N}+1\right)$th closest distance $∥S_{\left(k_{N}+1\right)}-s_{0}∥$; let $B_{s_{0}}=\{s:\;∥s-s_{0}∥\leq d_{k_{N}+1}\}$. Conditional on $d_{\left(k_{N}+1\right)}$, the tuples $\left(\theta_{\left(1\right)},S_{\left(1\right)}\right),\ldots ,\left(\theta_{\left(k_{N}\right)},S_{\left(k_{N}\right)}\right)$ are an ordered sample drawn according to probability density
(34)$$p\left(s,\theta |s_{0}\right)\propto 1_{B_{s_{0}}}p\left(s|\theta \right)\pi \left(\theta \right).$$

Conditionally on $d_{\left(k_{N}+1\right)}$, the accepted θ’s are independent, identically distributed (iid) realizations of the probability density
(35)$$\pi_{ABC}\left(\theta |s_{0}\right)=\frac{\pi \left(\theta \right)\int_{B_{s_{0}}}p\left(s|\theta \right)ds}{\int_{R^{p}}\pi \left(\theta \right)\int_{B_{s_{0}}}p\left(s|\theta \right)dsd\theta }.$$

#### 9.2.3. ABC MCMC Samplers

Again, we begin with a likelihood-free, MCMC algorithm whose stationary distribution exactly recovers the posterior, as motivation for the subsequent approximate samplers. Algorithm 4 can be seen as analogous to Algorithm 2.

**Algorithm 4:** Exact MCMC Sampler (Algorithm F in [181]).

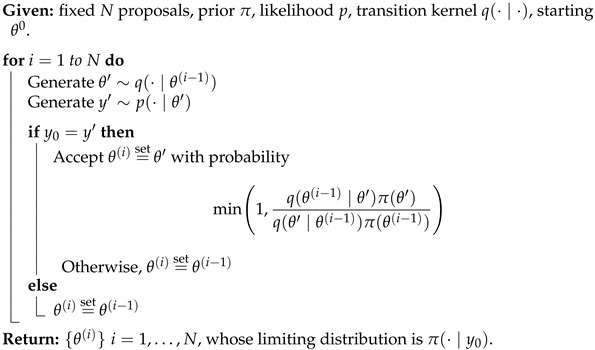



As before, the equality condition $y_{0}\equiv y^{\prime }$ remains practically unrealistic. One ABC-based approach removes this requirement in the following manner. Instead, *J*-many values $y_{j}^{\prime }$ are generated for each $\theta^{\prime }$, and the acceptance probability is weighted by the number of simulated $y_{j}^{\prime }$ falling within an ε-ball $B_{y_{0}}$ of the observed data. For the sake of brevity, this paper excludes more intricate ABC algorithms that address many complexities of Bayesian computations. Nevertheless, the development of ABC algorithms remains an open area of research.

### 9.3. ABC Strengths and Weaknesses

The obvious benefit of ABC to practitioners is the ability to make (approximate) inferences on previously inaccessible models. This occurs much in the same way that Markov Chain Monte Carlo methods broadened the class of models for which Bayesian analysis was possible. These methods are being applied in fields as disparate as astrophysics [184], geology [185], and others. The “forward” nature of ABC algorithms mirrors the design of many practical problems. Atmospheric, hydrological and pharmacokinetic models are often designed to run in a similar forward manner given a set of input parameters (take, for example, [186,187,188]).

ABC also possesses several attractive features in its implementation. First, ABC, as implemented in Algorithm 3 (kNN or fixed cutoff), is straightforward to code and understand. More importantly, these algorithms are also embarrassingly parallelizable. Since there is little computational overhead in administering the ABC sampling task across cores, we typically see speedups commensurate with the number of parallel operations. Section 9.4.2 provides a practical demonstration of this, where a sampling task is repeated across 1, 4, 8 and 16 cores.

ABC’s weakness is, to its credit, in the name: it is an approximation. Using it for inference should be tempered with an understanding that its output is a blend of the true posterior and the prior. Practitioners should be cognizant of this when selecting priors, as they influence results to a greater degree than is typically experienced in other methods such as MCMC or empirical Bayesian computations.

Moreover, the choice of summary statistics still requires attention and problem-specific knowledge. Section 9.5 highlights the different results that practical applications can produce based on disparate summary statistics, all crafted using expert knowledge. At the current stage of ABC development, it might be best to perform inference using several sets of summary statistics and make conclusions on the plurality of their results.

### 9.4. Examples

By considering ABC’s performance on a model with a known analytic solution, we can demonstrate some of the convergence properties outlined in Section 9.2.

#### 9.4.1. Example: Normal-Normal Hierarchical Model

Consider a normal-normal hierarchical model with an unknown mean ϑ and variance $\sigma^{2}$ under a conjugate Normal-Inverse-Gamma (NIG) prior. Assume we have *n* iid observations $y_{0}=\left[y_{1},\cdots ,y_{n}\right]$ from the model
(36)$$\begin{array}{cc} Y_{i}|\vartheta ,\sigma^{2} & ∼N(\vartheta ,\sigma^{2})\\ \vartheta ,\sigma^{2} & ∼NIG\left(\mu ,\lambda ,\alpha ,\beta \right)\;i.e.\;\;\vartheta |\sigma^{2}∼N\left(\mu ,\frac{\sigma^{2}}{\lambda }\right)\;\mathrm{and}\;\;\sigma^{2}∼\mathrm{IG}\left(\alpha ,\beta \right). \end{array}$$ Then, we have the following posterior distribution
(37)$$\vartheta ,\sigma^{2}|y_{0}∼\mathrm{NIG}\left(\mu^{*},\lambda^{*},\alpha^{*},\beta^{*}\right)$$
where
$$\mu^{*}=\frac{\lambda \mu +n\bar{y_{0}}}{\lambda +n};\;\lambda^{*}=\lambda +n;$$
$$\alpha^{*}=\alpha +\frac{n}{2};\;\beta^{*}=\beta +\frac{1}{2}\sum {\left(y_{i}-\bar{y_{0}}\right)}^{2}+\frac{n\lambda }{2(n+\lambda )}{\left(\bar{y_{0}}-\mu \right)}^{2}.$$

With this analytic solution, we may investigate the quality of results produced by ABC. Aspects of the problem itself, as well as implementation decisions made by the analyst, affect the final quality. For example, the problem may dictate the nature and number of data available, but we may choose the algorithm, tolerance and runtimes of our investigation.

To demonstrate this, consider estimating the posterior mean of ϑ (given by $\mu^{*}$ in (Equation 37)) having observed n=25 values from the normal–normal model where $\left(\vartheta_{0},\sigma_{0}^{2}\right)=\left(10,1\right)$. Typical Bayesian analysis requires soliciting a prior distribution as does ABC; we consider the NIG conjugate prior with all combinations of the following hyperparameters: μ∈{−10,0,10} to represent “poor”, “neutral” and “good” prior knowledge, λ∈{0.01,1} for agnostic and specific prior knowledge of ϑ (that is, a narrow or tight prior on ϑ for a given $\sigma^{2}$), and (α,β)∈{(1,1),(3,1)} for similarly broad and narrow prior beliefs concerning $\sigma^{2}$. ABC additionally requires specifying a sampling algorithm, distance metric and summary statistic. We employ the following distance metric and summary statistic combinations:1.Euclidean distance between summary $S\left(y\right)=\bar{y}$, i.e.,
$$\rho (S\left(y_{0}\right),S\left(y^{\prime }\right)=∥{\bar{y}}_{0}-{\bar{y}}^{\prime }{∥}_{2}.$$2.Euclidean distance between sufficient statistic: $S\left(y\right)={\left[\bar{y}\;\;\frac{1}{n}\sum {\left(y_{i}-\bar{y}\right)}^{2}\right]}^{⊤}$.3.Wasserstein-1 distance between identity statistic: S(y)=y. This is implemented in terms of the empirical distribution function $F_{y}\left(t\right)=\sum_{i=1}^{n}w_{i}^{\left(y\right)}1_{\left\{y_{i}\leq t\right\}}$ with normalizing weights $w_{i}^{\left(y\right)}$, and the corresponding distribution $F_{y_{0}}\left(t\right)=\sum_{j=1}^{n}w_{j}^{\left(y_{0}\right)}1_{\left\{b_{j}\leq t\right\}}$ for $y_{0}$. The Wasserstein-1 distance is given by
$$W_{1}\left(F_{y},F_{y_{0}}\right)=\int_{-\infty }^{\infty }\left|F_{y}\left(t\right)-F_{y_{0}}\left(t\right)\right|dt$$
and implemented with transport [189].4.K-L divergence between identity statistic S(y)=y, i.e.,
$$D_{\mathrm{KL}}\left(p_{y_{0}}∥p_{y}\right)=\int_{-\infty }^{\infty }p_{y_{0}}\left(t\right)\log\left(\frac{p_{y_{0}}\left(t\right)}{p_{y}\left(t\right)}\right)dt$$
and implemented with FNN [190].

While this model affords a sufficient summary statistic, this is rarely the case in situations where ABC is employed. We take *S* as the first moment of the sample participates in the common practice of using first, second, third, etc., ordered moments when constructing *S*. Here, we see how doing so may capture incomplete information about the sample. For such low-dimensional summaries, the Euclidean distance is common. Alternatively, the Wasserstein distance and K-L divergence (though not technically a distance) operate on the whole of the sample.

Next, a practitioner must choose a sampling algorithm. Algorithm 3 requires a fixed ε. A single value is unlikely to produce comparable results across the different choices of ρ given above due to their varying scales. Instead, we base our estimates on the M=100 best samples returned by an ABC sampler run for 1 min (15 s across 4 cores). This kNN flavor of ABC is typically the most user-friendly. Discrepancies in the amount of time each distance metric takes to compute, as well as the stochastic nature of generating ABC samples, means M=100 amounts to a slightly different ε for each simulation. However, all schemes generated ≈34,600 samples (min 32,546, max 35,955).

Inspecting Figure 8 in all cases, a “good” knowledgeable choice of μ=10 results in an unsurprisingly low error of estimation. However, the right-hand column demonstrates the perils of a narrow, specific prior. There is up to a 100-fold increase in error between λ=0.01 and λ=1. The choice of hyperparameters is far more influential than the choice of distance metric and summary statistic. However, the samples retained by the Wasserstein sampler do not estimate the posterior variance of ϑ as well as other choices. Its overestimation would lead to conservative claims and wide posterior density estimates.

With the effects of these modeling decisions in mind, we next consider individually the three sources of error inherent in ABC:1.Approximation Error: use of ε>0 in acceptance criteria.2.Information Loss: use of a non-sufficient summary statistic.3.Monte Carlo Error: estimating quantities using finite ABC samples.

While the third source of error is present in most any sampler-based methodology, the first two pertain specifically to ABC.

**Approximation Error:** We investigate the ABC estimates of E[ϑ∣y] and $E[\sigma^{2}|y]$ over a range of ε values. The normal–normal hierarchical model is still employed, again under an NIG(0,0.01,1,1) prior. We use the Euclidean distance between values of the sufficient statistic $s\left(y\right)=\left(\bar{y},\;\frac{1}{n}\sum^{n}{\left(y_{i}-\bar{y}\right)}^{2}\right)$ as the measure of discrepancy. Figure 9 and Figure 10 presents the mean absolute error of these estimates averaged over 100 replications. In all cases, posterior estimation improves as ε→0.

Other authors have also used this analytically tractable example to demonstrate their results [179]. While Figure 9 and Figure 10 may suggest choosing ε to be as small as possible, these figures belie the increased computational burden that reducing ε requires. This highlights the challenge practitioners face in choosing ε. The results in the left panel of Figure 11 display the error in estimating the posterior mean of ϑ under a fixed computation budget using samplers with increasing ε values, and their errors are compared, matching corresponding studies of [179]. The line follows the mean error over 100 replications; vertical bars indicate one standard deviation. Note that shrinking ε beyond a certain point produces too few accepted samples for reliable estimation, whereas large values accept many poor proposals. The ideal tolerance balances these sources of error. Thankfully, the expansive [179] provides theoretical insight into the conditions when an optimal ε can be found. Unfortunately, their result requires a sufficient statistic for the data and its joint likelihood with a model’s parameters, which are unlikely to exist in situations where ABC is employed. They suggest sampling until the number of accepted samples is proportional to $\varepsilon^{-4}$, which may help guide practitioners as to a tolerance value they should consider.

**Information Loss:** Consider the task of estimating the posterior marginal mean of ϑ and $\sigma^{2}$. We do so using samples from $\pi_{ABC}$ based on both the sufficient statistic $S_{1}\left(y\right):=\left(\bar{y},\frac{1}{n}\sum^{2}{\left(y_{i}-\bar{y}\right)}^{2}\right)$, as well as the (not sufficient) summary statistic $S_{2}\left(y\right):=\bar{y}$. This process is repeated 100 times to produce the mean error estimates seen in Table 3.

**Monte Carlo Error:** Finally, ABC ultimately produces samples. Thus, using them in estimation will incur sampling errors. Below, we fix ε=0.05, use a sufficient statistic in our sampler, and again estimate the posterior mean of ϑ based on M∈{100,200,⋯,1000} samples. This process is repeated 100 times. The results superficially mirror those seen in Figure 9 for shrinking ε→0; the mean absolute error falls with growing *M*. However, since ε=0.05 is fixed, we do not expect the error to shrink beyond the bias inherent in any ABC sample with ε>0. As an illustration, the middle panel of Figure 11 depicts the mean absolute error of ABC estimates of $E[\vartheta |y_{0}]$ based on *M* samples. The mean runtime (in seconds) given is depicted in red.

#### 9.4.2. Example: ABC Parallelization

ABC is infinitely parallelizable, an advantage not possessed by many other samplers used in Bayesian computation (e.g., MCMC). The task of simulating data and accepting desirable parameters can be replicated across any number of available processors. This imparts a (roughly) multiplicative factor to the number of acceptable samples an algorithm can produce in a given time budget. Equivalently, this can be understood to mean additional cores reduce the required runtime to produce a given number of samples to a fraction of the single core time.

To demonstrate this, consider the normal model used above and the task of generating *M* samples via ABC. Any specific estimation task (e.g., estimate posterior moments) is unlikely to require significant compute time, and we merely generate the sample parameters in this example. We fix $n=10,\varepsilon =0.25,\theta_{0}=\left(10,2\right)$ and generate $M=\{1,2^{2},2^{4},2^{6},2^{8},2^{10},2^{12}\}$ split over 1, 4, 8, 16 cores. We repeat this 500 times and report the average amount of real-world time each task required. This is illustrated in the right panel of Figure 11, where we demonstrate how parallelizing ABC across multiple cores (1 orange, 4 green, 8 blue, 16 purple) affects runtimes in ($\log_{2}$ seconds). For small *M*, the additional overhead required to administer parallel tasks to several cores increased overall runtime. For larger simulations, parallelization reduced runtime, as expected (as per [191]). This figure also demonstrates that more acceptable samples can be produced in a set runtime with the inclusion of more parallel processes for large tasks. Table 4 provides the untransformed runtimes (in seconds).

### 9.5. Example: Population Genetics and Ancestral Inference

The following application of ABC to ancestral inference reflects one of the earliest examples of ABC in practice [192,193]. Instead of exhaustively enumerating every possible ancestry to assess the likelihood, ABC allows practitioners to make inference after having simulated a large number of likely scenarios. Population geneticists often want to draw inference on parameters governing the genealogical history of a population. Specifically, they may aim to understand the genetic variation given a sample of DNA sequences. Recall that any given gene may come in one of several varieties (termed alleles), and a sample of genetic code from a population would show natural variability between subjects due to mutation. Commonly, researchers estimate the mutation rate and the time when their lineages first coalesce into a single common ancestor. A typical modeling framework is the Kingman’s coalescent, which assumes the observed genetic sequences $y_{0}$ can be modeled in terms of several meaningful parameters of interest θ, as well as the nuisance parameter *H*, the unobserved genealogical history. Here, $\theta =(\alpha ,\vartheta_{0})$ where α is the population’s growth rate and $\vartheta_{0}$ is its mutation rate, and we have the following likelihood:(38)$$p\left(y_{0}|\theta \right)=\int_{H}p\left(y_{0}|H\right)p\left(H|\theta \right)dH.$$ We forgo introducing the form the likelihood takes for different parameterizations of *H* for simplicity and merely note that the analytic solution does not exist. We are, however, able to generate sample DNA sequences under this model, choosing as our specific implementation the ms software by [194] with true parameters $\vartheta_{0}=50,\alpha =30$ and n=20 observations.

We estimated the posterior distribution of $\left(\alpha ,\vartheta_{0}\right)$ with the kNN flavor of ABC based on 1,000,000 proposals with 1000 retained. The priors on $\left(\alpha ,\vartheta_{0}\right)$ were independent U(0,200). The four summary statistics we considered were the average Hamming distance between pairwise sequences, the number of segregation sites (number of alleles), Tajima’s *D* and Fay-Wu’s $H_{0}$ (as a vector in $R^{2}$), and finally, the combination of all four of these (as a vector in $R^{4}$). These bespoke statistics represent a typical situation in which expert knowledge is employed. In all cases, we used the Euclidean distance between the summary statistics for the observed data $s_{0}$ and simulated data to determine the *k* nearest neighbors of $s_{0}$. Contours of the retained samples are presented in Figure 12.

It has been shown that ABC results compare favorably with those derived by sampling the posterior by means of an importance sampling algorithm [180]. The latter represents a noisy estimate of the true, inaccessible posterior. They remark that the importance samples are less influenced by the priors and more concentrated around the true data-generating model parameters. The average pairwise Hamming distance is a direct estimator of $\vartheta_{0}$, and the number of segregation sites is influenced by the mutation rate and growth rate. Tajima’s *D* and Fay-Wu’s $H_{0}$ statistics combine information on $\vartheta_{0}$ and the number of segregation sites. Somewhat surprisingly, the results in panel a for the pairwise Hamming distance do not suggest that it can be used to estimate $\vartheta_{0}$. The combination of all four statistics best estimates the posterior of $\left(\alpha ,\vartheta_{0}\right)$, as noted in [180].

## 10. Conclusions

Earlier, the question of how modern techniques are brought to bear on our collective problems was asked. Specifically, “are we simply increasing our prediction windows by increasing resolution, or are we learning to leverage more data into our calculations better?” While this question still remains, this paper attempts to show in a few choice examples that the realization is perhaps different—that we not *only* have been increasing our fidelity in prediction, but at the same time, we have opened new avenues into what *exactly* we mean by prediction. The statistical inference community is providing predictions not necessarily based on the exact boundary conditions that systems exhibit, rather, they present ensemble predictions, i.e., that out of X systems considered, Y presented the following similar results. Further, over the last 70+ years of CSE success, we have expanded our understanding of what a data manifold is and how to manipulate/analyze and extract useful information from it.

This paper is an overview of some challenging and open problems in multi-scale physics and engineering, and some equally challenging and open questions in data sciences. Addressing the data science challenges will help advance their usage in physics and engineering and will in turn facilitate advancements in these disciplines. Needless to say, the exposition in this paper is limited in its scope due to space constraints and the authors’ own research interests and specializations. Nevertheless, we offer an overview and discussion on four topics in multi-scale physics and engineering and four topics in ML. There are numerous open challenges and research questions interspersed in this document, which should interest researchers far beyond the eight topics discussed herein.

Numerical examples are provided for one of the four ML topics, namely ABC, to illustrate the power and scope of the proposals of this paper. There, we study several properties of ABC, including the choice of prior, and the choice of sampling algorithm, distance metric, summary statistics, the choice of the tolerance parameter and the ABC sampling size, the effect of the sample size and noise variance on the performance of the ABC algorithm. We also report on the efficiencies that may be achieved with ABC using parallel computational techniques. A replication of a study on using ABC techniques in population genetics and ancestral inference is also reported with some novel computational techniques. Overall, we find that ABC algorithms can produce very reasonable approximations to complex Bayesian computations as long as the tolerance parameter is small enough. Using parallel computing is highly recommended, and greatly facilitates ABC.

## Figures and Tables

**Figure 1 entropy-24-01134-f001:**
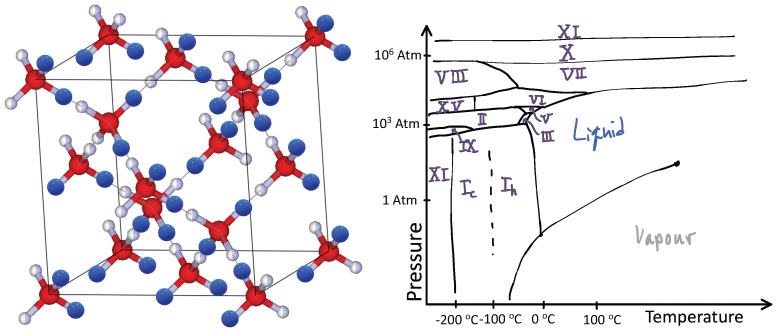
Water ice is not frozen. (**Left**): The crystal structure of Ice I${}_{c}$. Oxygen atoms (red) occupy the diamond lattice and hydrogen atoms (blue) occupy one of two possible (white) locations on the line between two oxygen atoms. Each oxygen atom obeys the “ice rules”: they must have two hydrogen atoms close to them and two far from them—a requirement imposed by the covalent bonding of 2H atoms to an O atom. (**Right**): Sketch of the complex phase diagram of water ice with pressure on a log scale and temperature on a linear scale (for more details, see Ref. [7]). Each ice phase is given a roman numeral. All of the higher temperature phases exhibit “proton disorder” with hydrogen atoms free to choose lattice sites as in ice I${}_{c}$.

**Figure 2 entropy-24-01134-f002:**
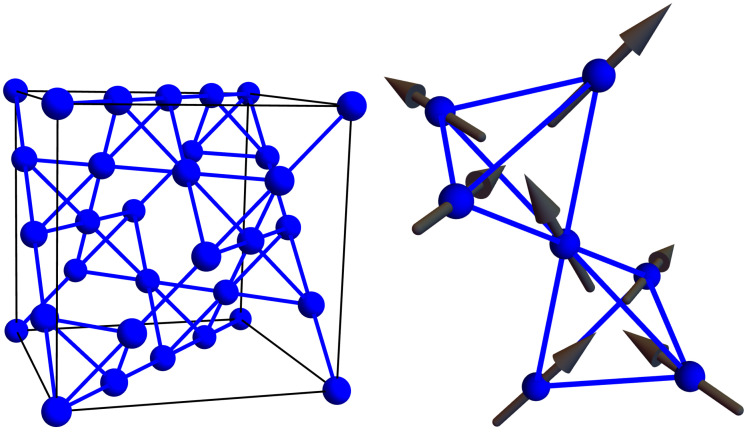
Spin ice is a magnetic ice. (**Left**): A cubic unit cell of the crystal Dy${}_{2}$Ti${}_{2}$O${}_{7}$ showing the positions of the Dy atoms as large blue spheres. The information needed to produce this figure can be found in the Dy${}_{2}$Ti${}_{2}$O${}_{7}$ .cif file at Materials Project [15]. (**Right**): Each Dy atom behaves like an atomic sized bar magnet, which we conventionally display as a vector, called a “spin vector”. The ice rules are obeyed in Dy${}_{2}$Ti${}_{2}$O${}_{7}$ at low temperatures where each tetrahedron replaces an oxygen atom, each spin replaces a hydrogen atom, and two spin vectors point inward to each tetrahedra and two point outward.

**Figure 3 entropy-24-01134-f003:**
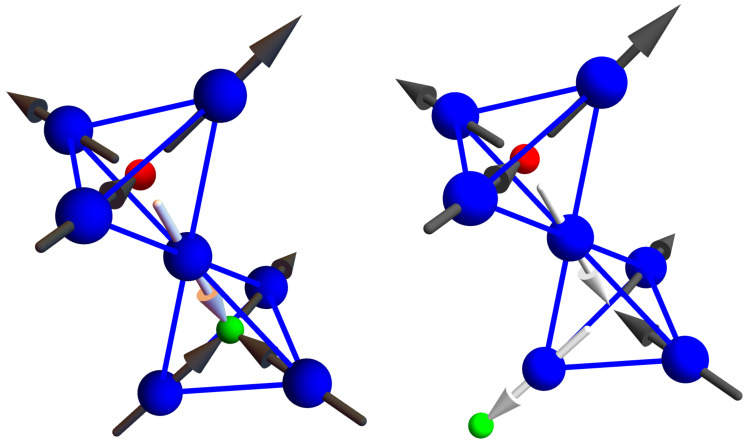
Spin flip process and monopole hopping. (**Left**): Starting from an ice-rule obeying spin configuration as in Figure 2, a pair of monopoles is formed after one spin flip as shown. Now, one tetrahedron has three spin vectors pointing out and one pointing in (a 3-in-1-out monopole) denoted by a red sphere, and one tetrahedron has on spin vector pointing out and three pointing in (a 1-in-3-out monopole) denoted by a green sphere. (**Right**): After a second spin flip, one of three things can happen: the monopoles annhilate and the two tetrahedra return to obeying the ice rules. Consequently, another pair of monopoles is created with one tetrahedra containing two (a 4-in-0-out or 0-in-4-out monopole, not shown), or more commonly, one of the tetrahedra returns to obeying the ice rules, and the monopole moves to another tetrahedra. This latter possibility leads to a gas of monopoles that move around the system much like electrons and holes do in semiconductors [20].

**Figure 4 entropy-24-01134-f004:**
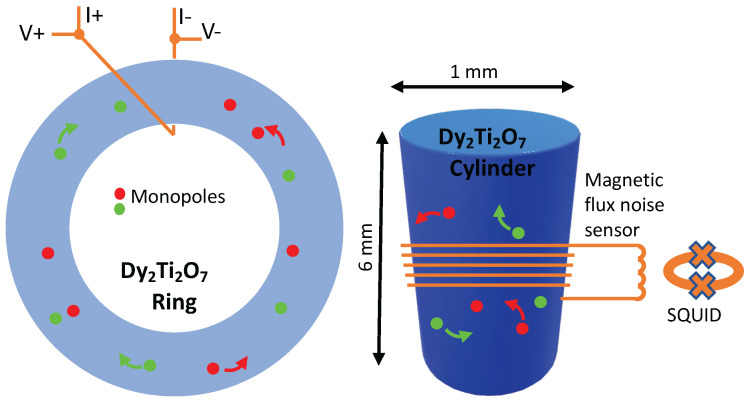
Spin ice rings and cylinders. Left: rings enable the study of monopoles as an electrically neutral fluid. The flow of monopoles around a ring enables the study [25] of the monopoles’ fluid properties; it is found that the monopoles behave as a supercooled liquid at low temperatures. Right: Noise detected in a coil wrapped around a cylinder of Dy${}_{2}$Ti${}_{2}$O${}_{7}$ enables a study of the random motion of the monopoles. The noise detected demonstrates this happens slowly, at audio frequencies (20–20,000 Hz) that are far below the expected MHz range of single spin-flip processes.

**Figure 5 entropy-24-01134-f005:**
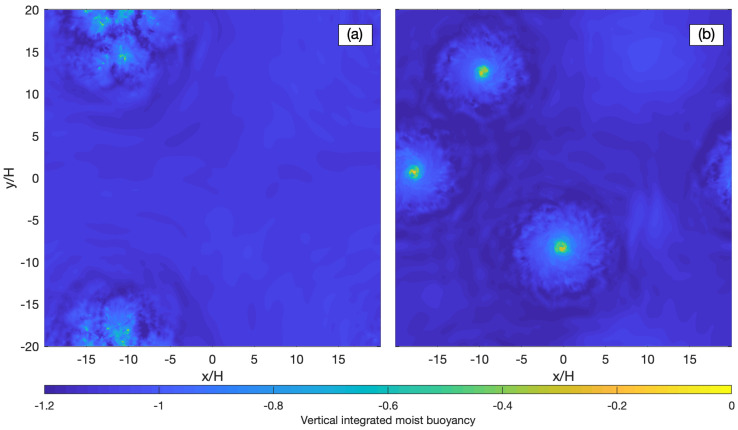
Instantaneous snapshots of vertically integrated moist buoyancy for the (**a**) no-rotation case and (**b**) rotation case in the A=40 domain, normalized by half the buoyancy difference between the upper and lower boundary. The normalized integrated buoyancy for an air column with the same moist buoyancy as the lower boundary is 0, and that for an air column with the same moist buoyancy as the upper boundary is −2.

**Figure 6 entropy-24-01134-f006:**
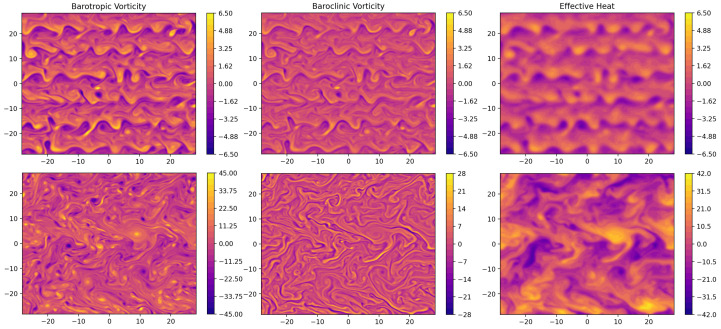
The barotropic and baroclinic vorticities, and the effective heat that acts as a source for the baroclinic vorticity. The top row is an entirely dry case, and as such, the effective heat is a proxy for temperature alone. The moist stratification in the dry case is defined as μ=1. The bottom is a moist case, so the effective heat is constructed from a combination of temperature and water vapor content. The moist stratification in this case is given by μ=4.0. The dry parameters are the same between the two cases. Note the different color scales—the dry case is much less energetic than the saturated.

**Figure 7 entropy-24-01134-f007:**
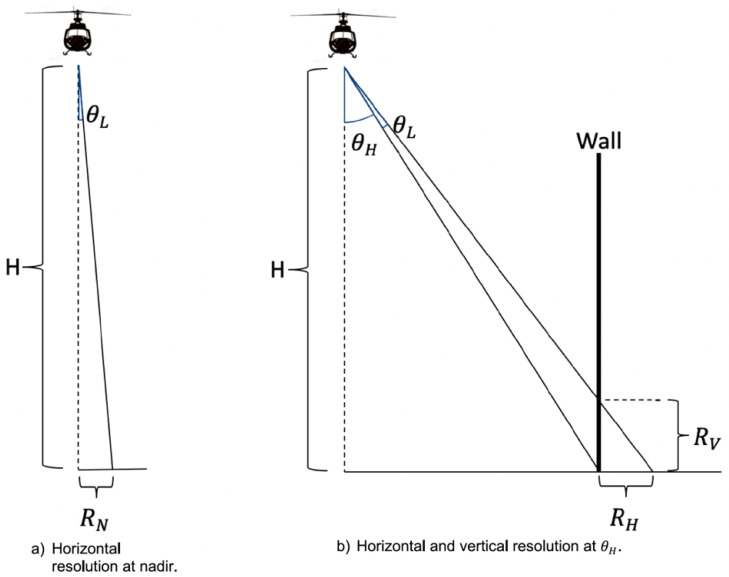
Geometric basis for predicting LiDAR data capture resolution from aerial platform (fixed wing or rotary), where $\theta_{H}$ is the data capture angle from nadir, and $\theta_{L}$ is the sensor beam spread.

**Figure 8 entropy-24-01134-f008:**
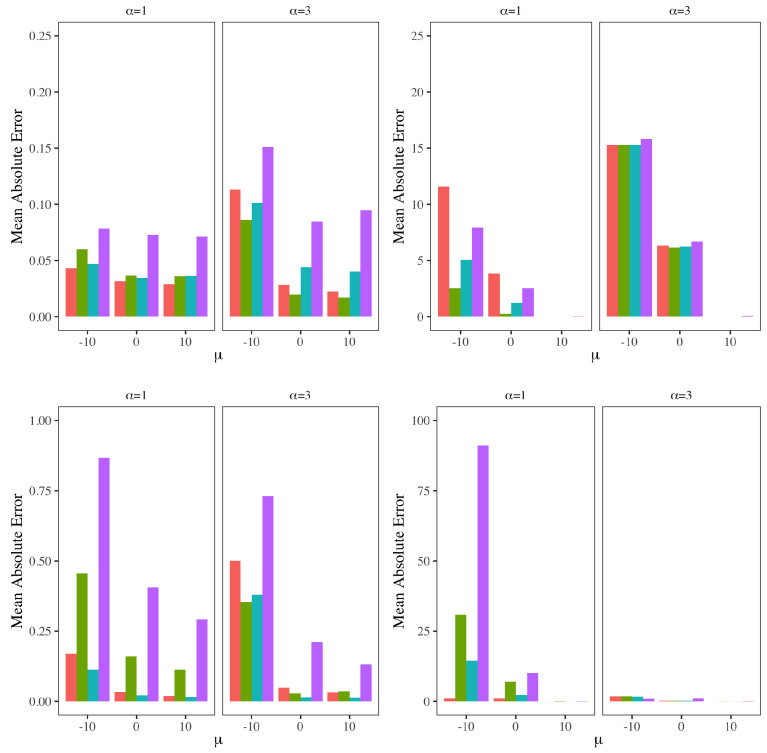
**Rows**: estimating posterior mean and variance of ϑ, top and bottom, respectively. **Columns**: prior hyperparameter λ=0.01,1. Distance metrics: sufficient Euclidean (orange), insufficient Euclidean (green), Wasserstein (blue), K-L divergence (purple).

**Figure 9 entropy-24-01134-f009:**
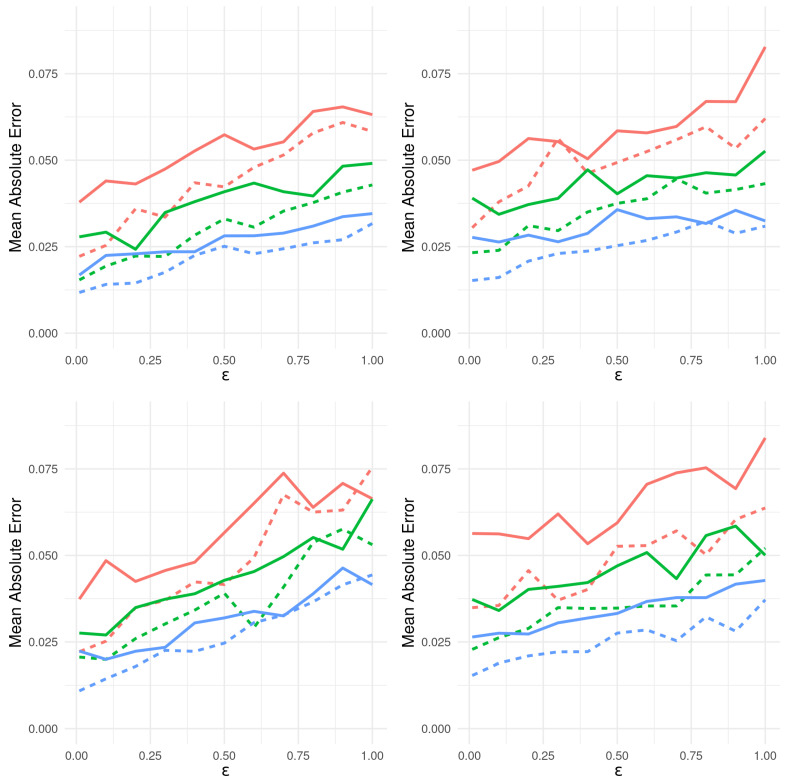
Estimating E[ϑ∣Y] with multiple configurations of Algorithm 3 for varying *n* (line type: n=10 solid, n=25 dashed), ε (horizontal axis), true data-generating model (facets clockwise from top left: $\left(\vartheta ,\sigma^{2}\right)=\left(1,1\right),\;\left(\vartheta ,\sigma^{2}\right)=\left(1,2\right),\;\left(\vartheta ,\sigma^{2}\right)=\left(10,1\right),\;\left(\vartheta ,\sigma^{2}\right)=\left(10,2\right)$), *M* (line color: M=50 orange, M=100 green, M=200 blue).

**Figure 10 entropy-24-01134-f010:**
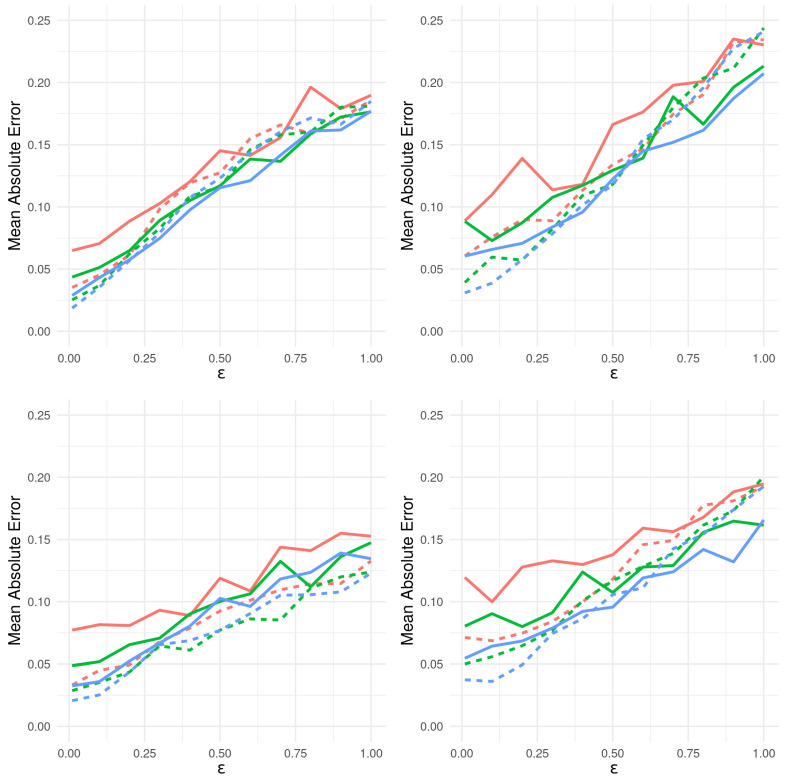
Estimating $E[\sigma^{2}|Y]$ with multiple configurations of Algorithm 3 for varying *n* (line type: n=10 solid, n=25 dashed), ε (horizontal axis), true data generating model (facets clockwise from top left: $\left(\vartheta ,\sigma^{2}\right)=\left(1,1\right),\;\left(\vartheta ,\sigma^{2}\right)=\left(1,2\right),\;\left(\vartheta ,\sigma^{2}\right)=\left(10,1\right),\;\left(\vartheta ,\sigma^{2}\right)=\left(10,2\right)$), *M* (line color: M=50 orange, M=100 green, M=200 blue).

**Figure 11 entropy-24-01134-f011:**
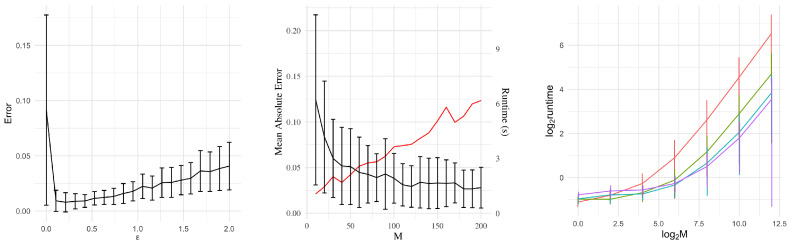
(**Left panel**): Estimation bias as a function of ε given a fixed computational budget. (**Middle panel**): Mean Absolute Error of ABC estimates of $E[\vartheta |y_{0}]$ based on *M* samples. Mean runtime (in seconds) given in red. (**Right panel**): Parallelizing ABC across multiple cores (1 orange, 4 green, 8 blue, 16 purple) affects runtimes ($\log_{2}$ seconds).

**Figure 12 entropy-24-01134-f012:**
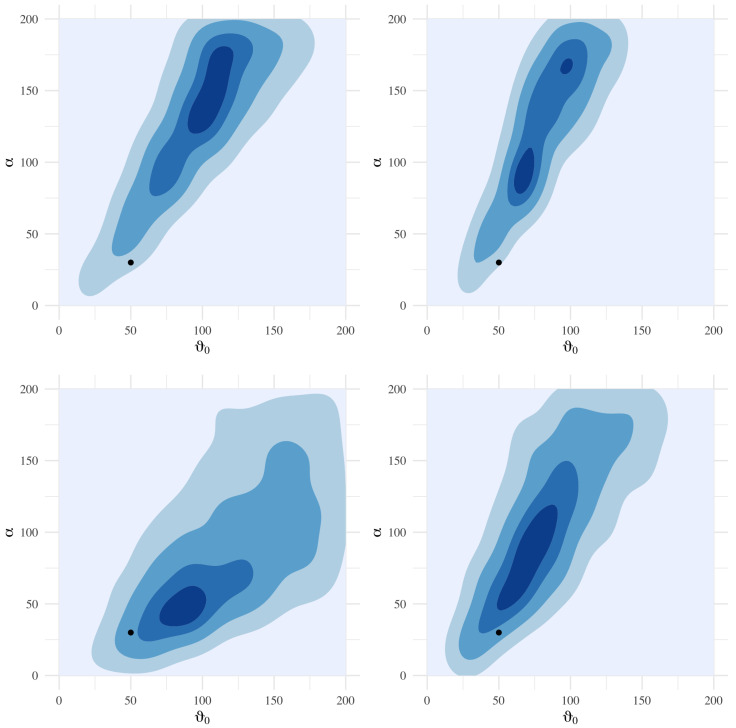
Posterior density estimates for $\left(\alpha ,\vartheta_{0}\right)$ under varying summary statistics (clockwise): average Hamming distance, number of segregation sites, Tajima’s *D* and Fay-Wu’s $H_{0}$, and all four statistics. Shading corresponds to {0.25,0.5,0.75,0.9} highest posterior density contours. True data-generating $\left(\alpha ,\vartheta_{0}\right)$ shown as black point in all frames.

**Table 1 entropy-24-01134-t001:** Spatiotemporal Generalized Techniques Comparison YES ≡ ✓, NO ≡ ✕. Many entries are not applicable, indicated by a dash, (-).

	Method Classification						Results Classification							
**Spatiotemporal Schemes**	**Linear/Non-Linear**	**Data/Eqn-Driven**	**Deterministic/Statistical**	**Continuous/Discrete**	**Boundary Conditions**	**Constraints/Conserved**	**Least Squares/Projection**	**Predictive**	**Generate PDF**	**Simulation**	**Scaling**	**Interpolation**	**Extrapolation**	**Quality/Limitations**
**(notation)**	**L/NL**	**D/E**	**D/S**	**C/D**			**LS/PR**							h=|Δr|,Δt
Particle Dynamical Systems	-	E	D	C	-	(E,p)	-	✓	∼	✓	✓	-	-	$P_{n}\left(\Delta t\right)$
Field Evolution	-	E	D	C	varied	(E,p)	PR	-	✓	✓	-	✓	∼	-
Lagranian Hamiltonian	-	E	D	C	varied	(E,p)	PR	✓	✓	✓	✓	-	-	-
PDE-Linear	L	E	D	C	varied	-	PR	✓	-	✓	-	✓	∼	-
PDE-Non-Linear	NL	E	D	C	varied	-	LS/PR	✓	-	✓	-	✓	∼	-
SDE [113]	NL	D	S	C	-	-	LS	partially	✓	✓	-	✓	✓	-
Finite Element Finite Difference [114]	L	-	-	-	per element	smooth	LS	-	-	-	-	✓	✓	$P_{n}\left(h\right)$
Forward Time Evolution	-	D/E	D	C/D	initial	rules	LS	✓	-	✓	-	-	✓	$P_{n}\left(iter\right)$
Markovian	L	D/E	D	C/D	initial	-	LS/PR	✓	∼	✓	-	✓	∼	$P_{n}\left(iter\right)$
Gaussian Process	-	D/E	S	C/D	-	asymp bound	LS	✓	✓	✓	✕	✓	✕	-
Machine Learning	NL	D	∼D+S	C+D	-	-	LS	mostly	✓/✕	∼	∼	✓	∼	N_nodes
Discrete Operations	L/NL	E	D	D	-	rules	LS	✓	-	✓	-	-	-	-
Modal Decomposition	L	D/E	-	C	-	ortho-gonality	PR	✕	-	-	✓	✓	✕	$N_{max}<\infty$
Virial Theorem	NL	E	D/S	C	-	✓	LS	-	-	-	-	-	-	-

**Table 2 entropy-24-01134-t002:** Specific Spatiotemporal Techniques Comparison YES ≡ ✓, NO ≡ ✕. Many entries are not applicable, indicated by a dash, (-).

	Schemes Employed													
**Specific Spatiotemporal Schemes**	**Particle/Dynamical**	**Field/Lattice Evolution **	**Lagr/Hamiltonian**	**PDE-Linear**	**PDE-Non-Linear**	**Stochastic DE**	**Finite Element/Diff.**	**Forward Time Evol.**	**Markov/Kernel**	**Modal Decomposition**	**Virial Theorem**	**Discrete Ops.**	**Gaussian Proc.**	**Neural Net**
Monte Carlo [118]	✕	✕	✓	-	-	-	✕	✕	-	-	-	-	-	-
Molecular Dynamics [119,120,121]	✓	✕	✓	-	-	✓	✓	✓	-	-	✓	-	-	-
Fluid Advection (CFD) [122]	✓	✕	✕	-	✓	-	✓	✓	✓	-	-	-	-	-
Ising Model [123]	-	✓	✓	-	-	-	-	✓	-	-	-	✓	-	-
Quantum Mechanics [112,124,125]	-	✓	✓	✓	✓	-	-	-	-	✓	-	-	-	-
Data Assimilation [126,127]	-	✓	✕	✓	✓	✓	-	-	-	-	-	-	-	-
Cellular Automata (Life) [107,108]	-	✓	✕	-	-	-	-	✓	-	-	-	✓	-	-
Kinetic Monte Carlo [128,129]	-	✕	-	-	-	-	-	-	-	-	-	-	-	✓
LiDAR [130]	-	✕	-	✕	✕	-	✓	-	✓	-	-	✓	-	✓
Spin Ice [13,20]	-	✓	✓	-	-	-	-	-	-	-	-	✓	-	✓
Rayliegh-Bernard Moist Conv. [71]	-	✓	-	-	✓	-	-	✓	-	-	-	-	✓	✓
DMD [109,110]	-	✓	✕	-	-	-	-	-	-	✓	-	-	-	-
Physics Informed NN [111]	-	✕	✕	-	-	-	-	✓	-	-	-	-	-	✓
Ocean Modeling - POM/ROM [131]	-	✓	-	✕	✓	✓	-	✓	-	-	-	-	-	-
Climate Modeling [132]	-	✓	-	✕	✓	✓	-	✓	-	-	-	-	-	-
Galactic Structure [133]	✓	✕	-	-	-	-	-	-	-	-	✓	-	-	-
Stock Futures [134]	-	-	-	-	-	✓	✓	✓	-	-	-	✓	✓	✓
Compressive Sensing [135]	-	-	-	-	-	✓	-	-	✓	-	-	✓	-	-

**Table 3 entropy-24-01134-t003:** $S_{2}$ does not contain any information on the variance of the generated data, and the estimates concerning $\sigma^{2}$ correspondingly suffer.

	$E_{\pi_{\mathrm{ABC}}}\left[\vartheta |Y\right]$ Error		$E_{\pi_{\mathrm{ABC}}}\left[\sigma^{2}|Y\right]$ Error	
ε	$S_{1}$	$S_{2}$	$S_{1}$	$S_{2}$
0.01	0.033 (0.029)	0.040 (0.036)	0.069 (0.074)	1.269 (1.673)
0.1	0.035 (0.031)	0.043 (0.038)	0.069 (0.064)	1.233 (1.937)
0.2	0.038 (0.032)	0.045 (0.040)	0.085 (0.074)	1.118 (1.188)
0.3	0.040 (0.033)	0.048 (0.052)	0.093 (0.071)	1.344 (4.544)
0.4	0.042 (0.036)	0.051 (0.042)	0.105 (0.078)	1.134 (1.327)
0.5	0.045 (0.037)	0.053 (0.045)	0.121 (0.087)	1.131 (0.995)

**Table 4 entropy-24-01134-t004:** In seconds, mean (sd) runtimes across 500 replications.

M	1 Core	4 Cores	8 Cores	16 Cores
1	0.46 (0.07)	0.51 (0.05)	0.51 (0.08)	0.58 (0.05)
$2^{2}$	0.57 (0.10)	0.51 (0.07)	0.58 (0.14)	0.65 (0.12)
$2^{4}$	0.83 (0.30)	0.62 (0.16)	0.60 (0.10)	0.68 (0.10)
$2^{6}$	1.88 (1.36)	0.92 (0.38)	0.79 (0.27)	0.83 (0.20)
$2^{8}$	6.15 (5.13)	2.25 (1.47)	1.58 (1.01)	1.42 (0.68)
$2^{10}$	23.56 (20.03)	7.48 (5.98)	4.16 (3.07)	3.46 (2.29)
$2^{12}$	93.00 (74.08)	26.10 (23.15)	14.35 (11.15)	11.66 (11.26)

## Data Availability

We have used publicly available datasets in this paper.

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
