# Peer review of "Machine Learning Methods for Multiscale Physics and Urban Engineering Problems"

_entropy, 2022, doi:10.3390/e24081134_

Round 1

Reviewer 1 Report

The use of the English language is entirely inappropriate and unacceptable for a submission to a prestigious academic Journal. The text requires crucial refinement and importantly vast professional proofreading by a native English speaker/proof-reader. This issue per se, renders desk-rejection. Indicatively, even in the title of the paper, the syntactic and grammatical errors are obvious “Machine learning methods for multiscale mixed resolutions problems”. The topic “multi-scale” refers to resolution methods. Either the statement should be “multi-resolution methods”, or “time-scale resolution” or “time-scale approaches”. Additionally, “mixed resolution” is redundant, because under “multi-scale” or “time-scale” resolution methodologies, is identical to stating “mixed” as an adjective to resolution. So, either “mixed” should be removed, or “multi-scale” should be removed. Lastly, there is no such scientific term as “resolutions problems”. Whether the authors mean “multi-scale resolution problems”, or “mixed resolution problems”, or “time-scale multi-resolution problems” and so on and so forth, is not tractable on my side. The content of the study does not include any novelty. Broadly known ML techniques are employed, instead of new techniques. The authors claim they tackle “physics-related and ML-related parts …”, does not make any sense. What does ML-related parts problems mean? And how is it possible scientifically to juxtapose physics-related problems vis-à-vis ML-related? ML techniques are derived from Data Science and can be applier to physics-related problems! Thus, ML is a tool, Physics is a scientific topical field (indeed a very wide one). Many parts of this study are incomprehensible to me, mostly due to the use of the English language.

Author Response

We thank the reviewer for their comments. We are concerned that the reviewer found the English in our paper to be inappropriate, and took several steps to ensure that the revised manuscript has satisfactory English and has a much clearer exposition of our ideas. To this end, we performed the following major revisions:

  1. The paper was first very carefully read and edited for grammar, stylistic issues and content by four co-authors whose first language is English. Then, we employed a software to further correct any remaining linguistic, spelling and grammatical issues. Finally, we employed professionals from a Center for Technical Writing help group in a US-based university, for further editorial suggestions and help with the use of correct linguistic practices. Thus, the manuscript has gone through at least seven rounds of English editing to address the referee’s concern. We believe that these steps have improved greatly the presentation quality of our paper. Any remaining English language error is of course our responsibility, and we will be happy to address them.
  2. We are sorry that in the first version of the paper, the terms “multiscale” and “mixed resolution” were not clarified properly. We use the term “multiscale” just the way the referee has interpreted it, that is, to refer to physical phenomena occurring at different scales. We had used the term “mixed resolution” primarily to refer to the fact that the observed data may be available at different resolutions as well. To address the reviewer’s concerns about this matter, we have taken several steps:
    1. We have now revised the title of the paper and the term “mixed resolution” no longer appears.
    2. We have made significant changes in the abstract to remove any ambiguity over these matters.
    3. We have now added some discussion in the Introduction to fully clarify multiscale phenomena and mixed-resolution data issues. The bulk of page 2 of the revised manuscript is on clarifying these details.
    4. We have taken care to not mix up these terms or concepts elsewhere in the paper.
  3. We appreciate the referee’s concerns about novelty, and discussing both research questions from physics as well as the data sciences.
    1. First, we have made it clear both in the abstract as well as in the introduction that this is an overview paper that is directly related to the scope of the special issue on Spatiotemporal Prediction and Simulation Methods at the Nexus of Statistical Physics, Spatial Statistics and Machine Learning. Indeed, this paper squarely fits into the description of the special issue that can be found in the journal’s website (https://www.mdpi.com/journal/entropy/special_issues/spatiotemporal_prediction) where the call is “to explore inter-disciplinary predictive approaches for spatiotemporal systems which combine ideas from statistical physics, space–time statistics, as well as statistical and machine learning”.  We submit that is the main theme of our paper, where instead of concentrating on one narrowly focused physics problem with an equally narrowly focused machine learning technique to drive a possible solution, we have taken the broader view of providing an exposition into several physics and engineering problems and several leading data science techniques that may be coupled to generate physically meaningful and fundamentally transformative We could have taken the narrow view point and concentrated on one of the discussed physics problems and produced partial answers by some variety of machine learning techniques, thus producing yet another one of the many papers that report incremental advancements and are often forgotten quickly. Instead, we preferred to provide an overview to the vast array of open research questions that may be exciting to many researchers besides us. We hope the referee will appreciate this choice, as indeed the other reviewers and editors seem to have done.
    2. We humbly disagree with the referee’s comment that there is no novelty in the paper. Despite the fact that this is an overview article, we have provided new materials in Sections 9.4 and partially in 9.5 (pages 32-37).

We thank the referee once again for their comments, and we hope we have addressed their concerns.

Reviewer 2 Report

This manuscript highlights the necessity of exploration of the utility of the application of modern Bayesian and more generally speaking machine learning methods to complex multi-scale spatio-temporal phenomena in physics and engineering.

Several phenomena, including spin ice and moist atmosphere, are reviewed in sufficient detail and an extensive list of modeling techniques is provided and classified.

Overall the manuscript is worthy of publication as it connects problems and possible solutions and may serve as a beacon to researchers in corresponding domains.

In my view the manuscript could be improved in the following way before being published (unless I missed something entirely in which case perhaps the authors could clarify their intent):

It would be worthwhile to actualize the problems in physics and engineering by spelling out what the current modeling (numerical) challenge is and propose a method from the list of methods in the second section as a potential candidate with an emphasis on multi-scale resolution.

As it stands the connection between the first part and the second is too vague from my point of view.

Minor corrections

1. Fig. 1, right, phase diagram of ice: the scales are not well-defined. Visibly they are log scales. (redo the diagram)

2. It might be worthwhile to mention that 2-2 (ice) rules are can be interpreted in terms of covalent bonding of 2H atoms to the oxygen atom.

3. Bottom page 4. phrase :
    "From a lattice perspective, we can define the location of each Dy atom by identifying which primitive face centered cubic unit cell it belongs to (labeled by integers n 1 , n 2 , and n 3 ) and which Dy cite within this unit cell α ∈ { 1, 2, 3, 4 } (see for example Ashcroft and Mermin[16]). "

    cite -> site, the phrase lacks grammatical cogency. the reference to Ashcroft and Mermin is too broad

4. Phrase at line 439 does not appear to be grammatical.

Author Response

We thank the reviewer for their insightful comments, and for noting that our manuscript is worthy of publication. We also indeed hope that this paper may serve as a beacon to researchers in multiple domains. We address specific points from the reviewer’s comments below:

  1. Referee’s comment: It would be worthwhile to actualize the problems in physics and engineering by spelling out what the current modeling (numerical) challenge is and propose a method from the list of methods in the second section as a potential candidate with an emphasis on multi-scale resolution. We thank the reviewer for this excellent suggestion, and we have now implemented in several parts of the paper. Please see the discussions in pages 6, 7, 10, 11, 12, 16 and 19 of the revised manuscript. where we have discussed the current modeling/numerical challenges and briefly discussed how the methods from Sections 6-9 may be helpful. Moreover, we made some smaller changes in Sections 6-9 to relate those better to the physics and engineering problems discussed in Sections 2-5. Such changes can be found, in particular, in Sections 6 and 8. We hope these revisions will address the reviewer’s concerns.
  2. Referee’s comment: 1, right, phase diagram of ice: the scales are not well-defined. Visibly they are log scales. (redo the diagram). Thank you for the nice suggestion. We have now redone the diagram and we have updated the axes of the phase diagram to explicitly denote the scale and mention this in the caption of Fig. 1. 

  1. Referee’s comment: It might be worthwhile to mention that 2-2 (ice) rules are can be interpreted in terms of covalent bonding of 2H atoms to the oxygen atom. This is a great suggestion. We mention this now in the caption of figure 1. Indeed, it is this bonding that gives rise to the ice rules, an important detail in the discussion. 

  1. Referee’s comment: Bottom page 4. phrase :
     "From a lattice perspective, we can define the location of each Dy atom by identifying which primitive face centered cubic unit cell it belongs to (labeled by integers n 1 , n 2 , and n 3 ) and which Dy cite within this unit cell α
    { 1, 2, 3, 4 } (see for example Ashcroft and Mermin[16]). "

        cite -> site, the phrase lacks grammatical cogency. the reference to Ashcroft and Mermin is too broad:

We have now updated this discussion and have made it more explicit. We have also provided a more precise reference, and fixed the typo cite->site.

  1. Referee’s comment: Phrase at line 439 does not appear to be grammatical. Thank you for pointing this, we have now corrected this.

Reviewer 3 Report

This paper is a good addition to the current trend in exploring what potential machine learning could provide us, especially since so much data is now available. The four problems selected so that the authors can convey their methodology are very good examples of some of the most challenging scientific problems we are facing today.  I am fairly familiar with two of the problems and found the authors narrative thought provoking and well-presented. I found the narrative in the other two problems also accessible.  The list of references is one of the strengths of this paper.  Although the paper is quite long, I think it is well-worth the expenditure of time.  While there are so many review articles these days on ML and how game-changing this new find may be for non-computer scientists, I found this article refreshingly different,  based on the authors choice of problems and fresh look at how they view ML's role may be in the future.

Author Response

We thank the reviewer for their insightful comments, and for finding that our work is a good addition to the current trend in exploring what potential machine learning could provide us. We hope the revised manuscript will be satisfactory to them.